# Towards a better characterisation of deep-diving whales' distributions by using prey distribution model outputs?

Auriane Virgili[1]*, Laura Hedon[1], Matthieu Authier[1,2], Beatriz Calmettes[3], Diane Claridge[4], Tim Cole[5], Peter Corkeron[5], Ghislain Dorémus[1], Charlotte Dunn[4], Tim E. Dunn[6], Sophie Laran[1], Patrick Lehodey[3], Mark Lewis[5], Maite Louzao[7], Laura Mannocci[8], José Martínez-Cedeira[9], Pascal Monestiez[10,11], Debra Palka[5], Emeline Pettex[2,12], Jason J. Roberts[13], Leire Ruiz[14], Camilo Saavedra[15], M. Begoña Santos[15], Olivier Van Canneyt[1], José Antonio Vázquez Bonales[16], Vincent Ridoux[1,11]

1 Observatoire PELAGIS, UMS 3462 CNRS—La Rochelle Université, La Rochelle, France, 2 ADERA, Pessac Cedex, Pessac, France, 3 Space Oceanography Division, CLS, Ramonville, France, 4 Bahamas Marine Mammal Research Organisation, Marsh Harbour, Abaco, Bahamas, 5 Protected Species Branch, NOAA Fisheries Northeast Fisheries Science, Woods Hole, Massachusetts, United States of America, 6 Joint Nature Conservation Committee, Inverdee House, Aberdeen, United Kingdom, 7 AZTI, Marine Research, Basque Research and Technology Alliance (BRTA), Pasaia, Spain, 8 MARBEC, Univ Montpellier, CNRS, Ifremer, IRD, Sète, France, 9 CEMMA, Pontevedra, Spain, 10 BioSP, INRA, Avignon, France, 11 Centre d'Etudes Biologiques de Chizé - La Rochelle, UMR 7372 CNRS—La Rochelle Université, Villiers-en-Bois, France, 12 Cohabys—ADERA, La Rochelle Université, La Rochelle, France, 13 Marine Geospatial Ecology Laboratory, Duke University, Durham, North Carolina, United States of America, 14 AMBAR Elkartea Organisation, Bizkaia, Spain, 15 Instituto Español de Oceanografía, Centro Oceanográfico de Vigo, Vigo, Spain, 16 Alnilam Research and Conservation, Madrid, Spain

* auriane.virgili@univ-lr.fr

**Data Availability Statement:** The data underlying this study are available on the OBIS SEAMAP website (http://seamap.env.duke.edu). We confirm

## Abstract

In habitat modelling, environmental variables are assumed to be proxies of lower trophic levels distribution and by extension, of marine top predator distributions. More proximal variables, such as potential prey fields, could refine relationships between top predator distributions and their environment. *In situ* data on prey distributions are not available over large spatial scales but, a numerical model, the Spatial Ecosystem And POpulation DYnamics Model (SEAPODYM), provides simulations of the biomass and production of zooplankton and six functional groups of micronekton at the global scale. Here, we explored whether generalised additive models fitted to simulated prey distribution data better predicted deep-diver densities (here beaked whales *Ziphiidae* and sperm whales *Physeter macrocephalus*) than models fitted to environmental variables. We assessed whether the combination of environmental and prey distribution data would further improve model fit by comparing their explanatory power. For both taxa, results were suggestive of a preference for habitats associated with topographic features and thermal fronts but also for habitats with an extended euphotic zone and with large prey of the lower mesopelagic layer. For beaked whales, no SEAPODYM variable was selected in the best model that combined the two types of variables, possibly because SEAPODYM does not accurately simulate the organisms on which beaked whales feed on. For sperm whales, the increase model performance was only marginal. SEAPODYM outputs were at best weakly correlated with sightings of deep-diving

that authors had no special access to the
underlying data and all interested researchers can
obtain the data from the OBIS SEAMAP in the
same way that the authors obtained data by
contacting the data providers via the OBIS
SEAMAP website.

**Funding:** AV's doctoral research grant was funded
by the Direction Générale de l'Armement (DGA).
ML was funded by a Ramón y Cajal (RYC-2012-
09897) postdoctoral contract of the Spanish
Ministry of Economy, Industry and
Competitiveness. This study is a contribution to the
CHALLENGES (CTM2013- 47032-R) project of the
Spanish Ministry of Economy, Industry and
Competitiveness. MA and EP are affiliated to
ADERA a private business dedicated to providing
support for the management of research project in
public institutions in Region Nouvelle Aquitaine. In
this study, ADERA solely provided support in the
form of salaries for authors MA and EP, but did not
have any additional role in the study design, data
collection and analysis, decision to publish, or
preparation of the manuscript. The specific roles of
these authors are articulated in the 'author
contributions' section.

**Competing interests:** MA and EP are employed by
the commercial company ADERA which did not
play any role in this study beyond that of employer.
This does not alter our adherence to PLOS ONE
policies on sharing data and materials.

cetaceans, suggesting SEAPODYM may not accurately predict the prey fields of these taxa.
This study was a first investigation and mostly highlighted the importance of the physio-
graphic variables to understand mechanisms that influence the distribution of deep-diving
cetaceans. A more systematic use of SEAPODYM could allow to better define the limits of
its use and a development of the model that would simulate larger prey beyond 1,000 m
would probably better characterise the prey of deep-diving cetaceans.

## Introduction

Cetaceans are subject to many anthropogenic threats such as vessel collisions, pollutants,
bycatch and underwater noise, leading to the decline of many populations [1–3]. To develop
effective conservation strategies, describing and predicting their distribution is essential
because cetacean distributions, particularly deep-diving cetacean distributions, are barely
known due to data scarcity which is an important pitfall for conservation. Habitat modelling is
a useful tool to predict species distributions [4]. In these models, species sightings are related
to proximal or distal variables that are assumed to influence their distributions. Proximal vari-
ables are biological variables to which the species is assumed to react more directly (*e.g.* prey
distribution) than distal variables, which describe the physical environment [5]. Distal vari-
ables encompass two types of variables; physiographic variables which are static descriptors
that relate to the bathymetry (*e.g.* depth, slope) and oceanographic variables which are
dynamic predictors that describe the water masses (*e.g.* sea surface temperature—SST, eddy
kinetic energy—EKE, surface chlorophyll *a* concentration). These variables, obtained from sat-
ellite imagery or numerical models, are more widely available than proximal data and are
often used to describe and predict marine top predator distributions [6–8].

In the marine environment, the surface chlorophyll *a* concentration is commonly used as a
proxy for the biomass of phytoplankton in cetacean habitat models [8–10], but the distribution
and biomass of low- (phytoplankton and zooplankton) and mid- (micronekton) trophic levels
are more proximal predictors of top predator distribution [7, 11]. Surface chlorophyll *a* con-
centration data is widely used because it is remotely-sensed from satellites and readily available
at a global scale, but there is a time-lag between a change in phytoplankton biomass and its
effects on upper trophic levels [7, 9, 11–14]. The use of more proximal variables, such as prey
biomass, could reduce this lag because marine top predators are expected to be mostly sensi-
tive to prey abundance and because such micronekton outputs are closer to trophic level of
apex predator prey [11, 15–18]. However, the limited spatio-temporal extent of prey data avail-
able from *in situ* sampling is a major bottleneck for modelling predator distributions from the
distribution of their prey over large oceanic regions. Ecosystem models simulating the biomass
and production of low- and mid-trophic levels provide a new numerical way to fill the prey
data gap globally [19–22]. The Spatial Ecosystem And POpulation DYnamics Model (SEAPO-
DYM) provides simulations of the global 3-dimensional distributions of zooplankton and six
functional groups of the micronekton defined by their daily vertical migration patterns in
three biological layers from 0 to *ca.* 1,000 m. SEAPDOYM was initially developed to model
tuna populations for fishery applications [23, 24], but its usage was extended to predict turtles,
cetaceans and elephant seals habitat use [20–22, 25, 26].

In a previous study [27], we developed habitat models for deep-diving cetaceans (beaked
whales *Ziphiidae*, sperm whales *Physeteriidae* and *Kogia* whales *Kogiidae*) using physiographic
variables together with surface oceanographic variables. Deep-divers are species of interest

because they are sensitive to underwater noise pollution [28, 29] and an accurate knowledge of their distribution is crucial to mitigate the impact of human activities. As deep-diving cetaceans spend most of their time at depth and generally feed on meso- to bathypelagic prey (*e.g.* [30, 31]), the use of surface variables may limit the ability to correctly infer their habitats. Consequently, using the same sighting survey data as in Virgili et al. [27], here we aimed to explore (1) if models fitted to simulated prey data explained the beaked and sperm whale distributions better than models fitted to environmental variables only and (2) if the combination of environmental and simulated prey data would further improve model fit. We compared the fit and explanatory power of three Generalised Additive Models (GAMs), fitted to beaked and sperm whales sighting data and (1) to environmental variables (hereafter 'ENVIRONMENTAL model'), (2) to prey distribution variables simulated from SEAPODYM (hereafter 'SEAPODYM model', differing from 'SEAPODYM' alone that refers to the model of Lehodey et al. [19] that simulated the prey distribution data) and (3) to a combination of both variable types (hereafter 'COMBINED model'). We then predicted beaked and sperm whale densities in the North Atlantic Ocean and compared the three models. We hypothesised that SEAPODYM variables would better explain the distribution of beaked whales and sperm whales because they are more proximal variables and because they characterise the deeper layers in which deep-divers feed on. We also expected a better explanatory power of the model that combined the two types of variables because it considered static variables known to influence the beaked and sperm whale distributions (*e.g.* [32, 33]), together with prey distribution data, presumably better suited to describe the species distributions.

## Material and methods

### Study area

The study area encompassed the North Atlantic Ocean from the Guiana Plateau to Iceland, *i.e.* approximately from 5–65˚N, excluding semi-enclosed seas like the Gulf of Mexico, the Hudson Bay, the Baltic sea and the Mediterranean Sea (Fig 1A).

In the North Atlantic Ocean, the general thermohaline circulation is characterised by the formation of deep saline and cold-water masses flowing from the Labrador basin and Greenland Sea southward and warmer surface waters generally flowing northward but affected by a large gyre [34, 35]. This subtropical gyre is delimited by the North Atlantic, Azores and Canary Currents in the east and the North Equatorial Current and Gulf Stream in the west. The latter is narrower and swifter than its eastern counterparts and follows the continental slope from Florida to the Grand Banks off Newfoundland. Large seasonal variations of the wind field (except in the subtropical area), high salinity and a general decreasing gradient of temperature from west to east (about 8˚C difference) are characteristics of the North Atlantic Ocean central gyre [35]. Within this ocean basin, primary production is quite low in the tropical zone, and varies seasonally elsewhere. Maximum productivity is reported in winter in the subtropical zone; a spring bloom and oligotrophic summer conditions are found at mid-latitude; and productivity is maximum in summer in the subpolar zone [36].

### Data collection and collation

We used effort and sighting data assembled in Virgili et al. ([27]; Fig 1). We only considered beaked and sperm whale sightings and effort data recorded in the North Atlantic Ocean. As deep-divers are difficult to detect during surveys and the typically small number of sightings recorded during a single survey is insufficient to fit habitat models, we assembled visual shipboard and aerial surveys performed by nine independent organisations in the North Atlantic Ocean between 1998 and 2015 (details of the surveys in S1 Table in S1 Appendix). A single

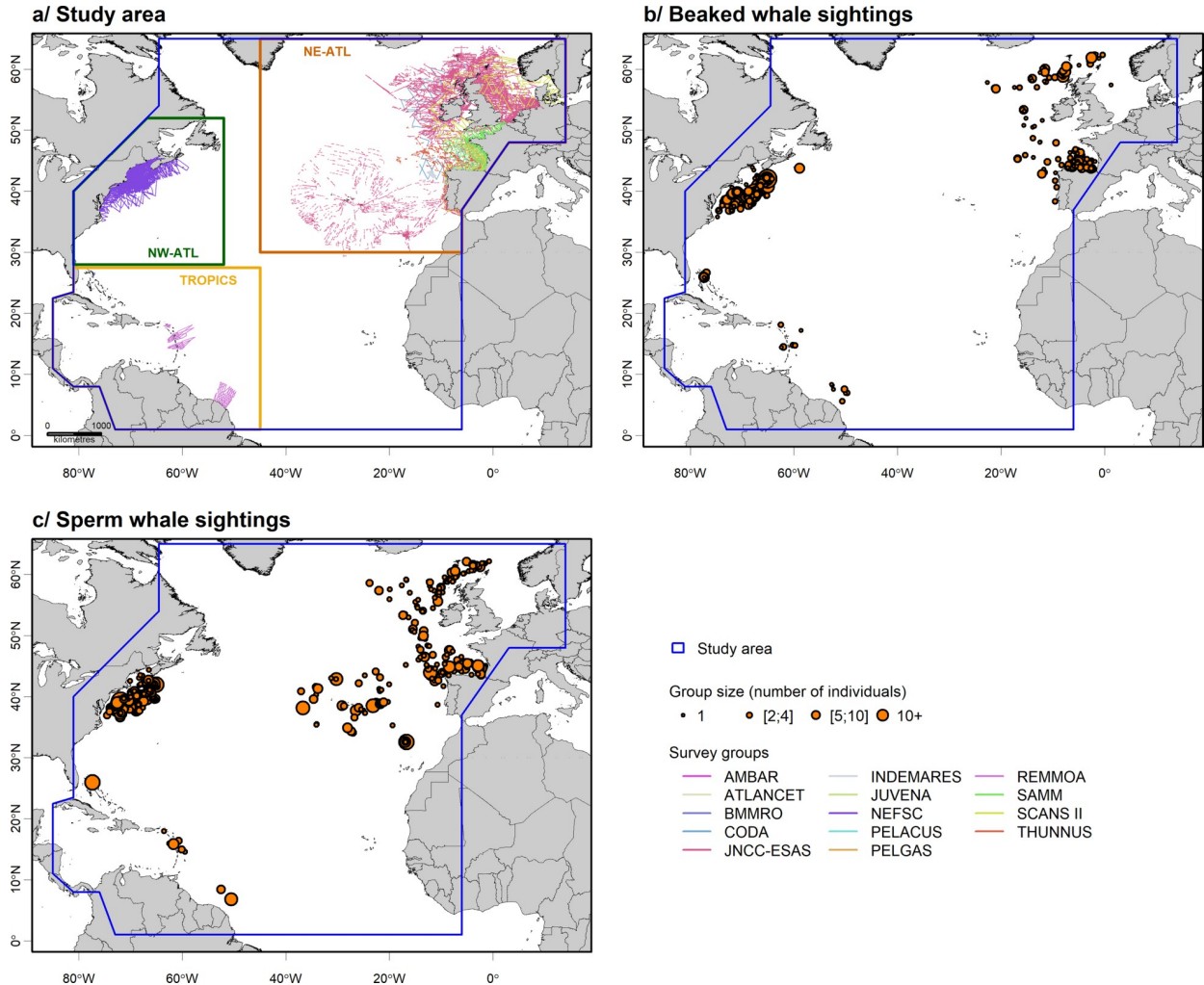

**Fig 1. Study area divided into sub-regions showing assembled survey effort (a), along with beaked whale (b) and sperm whale (c) sightings recorded during all surveys.** The blue polygon delineates overall study area and other polygons delineate sub-regions. Surveys were carried out along transects following a line-transect methodology (survey details in S1 Table in S1 Appendix). Sightings were classified by group sizes with each point representing one group of individuals and point size representing the number of animals in a group. Base map from https://www.gebco. net/.

common dataset was created, aggregating all survey datasets standardised for units and formats. Effort data were linearized and divided into 5 km segments using ArcGIS 10.3 [37] and the Marine Geospatial Ecology Tools software [38].

Except for the JNCC-ESAS surveys that used a 300-m strip-transect methodology, cetacean sightings were recorded following line-transect methodologies that allow Effective Strip Width (ESW) to be estimated from the measurement of the perpendicular distances to the sightings [39]. For each taxon, we built a hierarchical model to estimate the ESW depending on observation conditions and survey types (for more details, see [27]).

To account for the difficulty to identify them at species level, beaked whale species were pooled into one group including Cuvier's beaked whales (*Ziphius cavirostris*), mesoplodonts (*Mesoplodon* spp.) and northern bottlenose whales *(Hyperoodon ampullatus).*

A total of 481 sightings of beaked whales and 542 sightings of sperm whales, mainly distributed in the northeast and the northwest Atlantic Ocean (north of the 35˚N latitude) were

assembled for the present study (Fig 1B and 1C). Aggregated data represented about 1,101,200 km of on-effort transects (*i.e.* following a transect at a specified speed and altitude with a specified level of visual effort) of which 58% was carried out by plane and the rest by boat (Fig 1A and Table 1).

Moran's and Geary's indexes were calculated to ensure there was no spatial autocorrelation in the data using the 'spdep' R-package [40].

## Data processing

**Static and oceanographic variables.** We considered static and oceanographic variables that can affect the distribution of beaked and sperm whales (Table 2). All variables were resampled at a 0.25˚ resolution to match the spatial resolution of the SEAPODYM variables (Table 2 and S2.1-S2.3 Fig in S2 Appendix). Depth and slope were derived from GEBCO-08 30 arc-second database (https://www.gebco.net/; 30 arc-second is approximately equal to 0.008˚). The surface of canyons and seamounts per 0.25˚ cell was calculated in ArcGIS 10.3 from the shapefile of canyons and seamounts provided by Harris et al. [41]. Oceanographic variables, relating to the movements of water masses were upscaled at a monthly resolution, *i.e.* averaged over the 29 days prior to each sampled day to avoid gaps in remote sensing oceanographic variables and to consider a potential time-lag between an environmental condition and its effect on higher trophic levels. The mean, standard error and gradient of SST were calculated from the GHRSST Level 4 CMC SST v.2.0 ([42], https://podaac.jpl.nasa.gov/dataset/CMC0.2deg-CMC-L4-GLOB-v2.0). Spatial gradients of SST were calculated as the difference between the minimum and maximum SST values found in the eight pixels surrounding any given pixel of the grid. The Aviso ¼˚ DT-MADT geostrophic currents dataset was used to compute mean and standard deviation of Sea Surface Height (SSH) and EKE (https://www.aviso.altimetry.fr/en/data/products/sea-surface-height-products/global/madt-h-uv.html). Net primary production (NPP) was derived from SeaWIFS and Aqua using the Vertically Generalised Production Model (http://orca.science.oregonstate.edu/1080.by.21608day.hdf.vgpm.s.chl.a.sst.php) and was used as a proxy of prey availability.

**SEAPODYM variables.** SEAPODYM is a numerical model used to compute the spatial distribution of the biomass and the production of organisms, the micronekton and the zooplankton, in different layers of the water column [19, 23, 43]. For each functional

**Table 1. Effort performed by platform type and Beaufort sea-state per sector in the North Atlantic Ocean.**

| Sectors | Total survey effort (km and %) | Total aerial effort (km) | Total shipboard effort (km) | Total effort by Beaufort sea-state class (km) | | | | |
|---|---|---|---|---|---|---|---|---|
| | | | | 0–1 | 1–2 | 2–3 | 3–4 | 4–7 |
| NE-ATL | 521,000 | 70,400 | 450,600 | 76,700 | 118,500 | 135,700 | 190,000 | 200 |
| | 47% | | | | | | | |
| NW-ATL | 561,100 | 549,800 | 11,300 | 43,000 | 122,400 | 200,900 | 132,700 | 62,100 |
| | 51% | | | | | | | |
| TROPICS | 19,100 | 15,400 | 3,700 | 10,600 | 2,500 | 3,700 | 2,300 | 0 |
| | 2% | | | | | | | |
| STUDY AREA | 1,101,200 | 635,600 | 465,600 | 130,300 | 243,400 | 340,300 | 324,900 | 62,300 |
| | 58% | 42% | 11% | 22% | 31% | 30% | 6% |

This table presents the total effort conducted in each sector broken down by platform type and Beaufort sea-state. Beaufort sea-state values reported with decimals in the surveys were rounded up. For the analyses, all segments with Beaufort sea-state > 4 were excluded. 'NE-ATL' means northeast Atlantic Ocean and 'NW-ATL' means northwest Atlantic Ocean.

**Table 2. Candidate environmental and SEAPODYM predictors used for the habitat-based density modelling.**

| Variables used in the study with abbreviations and units | Original Resolution | S | Effects on pelagic ecosystems of potential interest to deep-divers |
|---|---|---|---|
| **Physiographic** | | | |
| Depth (m) | 30 arc sec | A | Deep-divers feed on squids and fish in the deep-water column |
| Slope (˚) | 30 arc sec | A | Associated with currents, high slopes induce prey aggregation or enhanced primary production |
| Surface of canyons and seamounts in a 0.25˚ cell–S.can.seam (km$^2$) | 30 arc sec | B | Deep-divers are often associated with canyons and seamounts structures; the variable indicates the proportion of this habitat in each cell |
| **Oceanographic** | | | |
| Mean sea surface temperature–SST (˚C) | 0.2˚, 1d | C | Variability over time and horizontal gradients of SST reveal front locations, potentially associated with prey aggregation or enhanced primary production |
| Mean gradient of SST–SSTgrad (˚C) | 0.2˚, 1d | C | |
| Mean of sea surface height–SSH (m) | 0.25˚, 1d | D | High SSH is associated with high mesoscale activity and enhanced prey aggregation or primary production |
| Mean of eddy kinetic energy–EKE (m$^2$.s$^{-2}$) | 0.25˚, 1d | D | High EKE relates to the development of eddies and sediment resuspension inducing prey aggregation |
| Mean of net primary production–NPP (mgC.m$^{-2}$.day$^{-1}$) | 9 km, 8d | E | Net primary production as a proxy of prey availability |
| **SEAPODYM variables** | | | |
| Euphotic depth–Euph. depth (m) | 0.25˚, 1w | F | Depth of the euphotic zone as proxy of prey availability |
| Epipelagic biomass and production–Epi. B. (g.m$^{-2}$) and Epi. P. (g.m$^{-2}$.day$^{-1}$) | 0.25˚, 1w | F | All these variables relate to the distribution of potential direct or indirect prey of deep-divers. |
| Non-migrant upper mesopelagic biomass and production–U. meso. B. (g.m$^{-2}$) and U.meso. P. (g.m$^{-2}$.day$^{-1}$) | 0.25˚, 1w | F | |
| Migrant upper mesopelagic biomass and production–M.U. meso. B. (g.m$^{-2}$) and M.U.meso. P. (g.m$^{-2}$.day$^{-1}$) | 0.25˚, 1w | F | |
| Non-migrant lower mesopelagic biomass and production–L. meso. B. (g.m$^{-2}$) and L.meso. P. (g.m$^{-2}$.day$^{-1}$) | 0.25˚, 1w | F | |
| Migrant lower mesopelagic biomass and production–M.L.meso. B. (g.m$^{-2}$) and M.L.meso. P. (g.m$^{-2}$.day$^{-1}$) | 0.25˚, 1w | F | |
| Highly migrant lower mesopelagic biomass and production–H. m.L.meso. B. (g.m$^{-2}$) and H.m.L.meso. P. (g.m$^{-2}$.day$^{-1}$) | 0.25˚, 1w | F | |
| Zooplankton biomass and production–Pk. B. (g.m$^{-2}$) and Pk. P. (g.m$^{-2}$.day$^{-1}$) | 0.25˚, 1w | F | |

All variables were resampled at a 0.25˚ resolution. S: Sources. A: https://www.gebco.net/; 30 arc-second is approximately equal to 0.083˚. B: Harris et al. [41]. C: Canada Meteorological Centre [42], https://podaac.jpl.nasa.gov/dataset/CMC0.2deg-CMC-L4-GLOB-v2.0. D: https://www.aviso.altimetry.fr/en/data/products/sea-surface-height-products/global/madt-h-uv.html. E: http://orca.science.oregonstate.edu/1080.by.2160.8day.hdf.vgpm.s.chl.a.sst.php. F: Lehodey et al. [19]. Although it is an oceanographic environmental variable, the euphotic depth was included in the SEAPODYM models because it is this variable that defines the three layers of SEAPODYM.1d: daily; 8d: eight days; 1w: weekly.

group (zooplankton and micronekton) biomass (in g.m$^{-2}$) and production (in g.m$^{-2}$.day$^{-1}$) are simulated at a 0.25˚ resolution. Production is defined as the recruitment of a new cohort of organisms into a micronekton functional group when they reach 1 g body mass (fixed value). Energy transfers from the primary production to the groups of micronekton are parameterised in the model and a system of advection–diffusion–reaction equations, that take into account the vertical migrations of organisms, are used to model recruitment, ageing, mortality, and passive transport with horizontal currents [19, 23, 43]. In SEAPODYM, the zooplankton is defined as all non-migratory phytoplanktivorous organisms with a size between 200 μm and 2 cm that live in the epipelagic layer [44]. Micronekton encompasses active swimming organisms in the range of 1–20 g and 2–20 cm and includes fishes, crustaceans and cephalopods [44, 45]. Depending on the vertical

distribution of the organism biomass; three layers are defined according to the euphotic depth (*i.e.* the layer of sea water that receives enough sunlight for photosynthesis to occur; Fig 2; [45]), computed according to the VGPM model (Vertically Generalized Production Model) of Behrenfeld &Falkowski [46]. The epipelagic layer extends from the surface to 1.5*euphotic depth; the upper mesopelagic layer, between 1.5 and 4.5*euphotic depth and the lower mesopelagic layer which extends from 4.5 to 10.5*euphotic depth with a maximum set at 1,000 m [45]. Micronekton can undertake nycthemeral vertical migrations between these three layers.

According to migration patterns, micronekton includes six functional groups: epipelagic, non-migrant upper mesopelagic, migrant upper mesopelagic, non-migrant lower mesopelagic, migrant lower mesopelagic and highly migrant lower mesopelagic organisms (Fig 2; [19, 23, 45]). These migrations are induced by daylight variations and may be due to a strategy of predator avoidance: during daytime, they dive in deeper waters to avoid predation [47]. Migrant upper mesopelagic and highly migrant lower mesopelagic organisms migrate between the epipelagic layer at night and respectively the upper mesopelagic and lower mesopelagic layers during daytime while the migrant lower mesopelagic organisms migrate between the upper mesopelagic and the lower mesopelagic layers. SEAPO-DYM makes it possible to obtain, in each grid cell, an estimate of the total quantity of organisms of each functional group present in each layer and of the productivity of the functional group in each layer (the greater the production, the richer the environment). A positive relationship of predators with production variables would indicate a relationship with the distribution of smaller and more abundant prey of the functional group while a positive relationship with biomass variables would indicate a relationship with the distribution of larger prey.

Contrary to oceanographic environmental variables, SEAPODYM variables were available at a weekly resolution *i.e.* averaged over the 7 days prior to each sampled day, and not upscaled at a monthly resolution to observe a potentially more direct link between prey and predator distribution. Due to the absence of the lower mesopelagic layer on the continental shelf, a zero, and not an NA (for not available), was assigned to the effort segments on the continental shelf to avoid deleting the segments during model fitting.

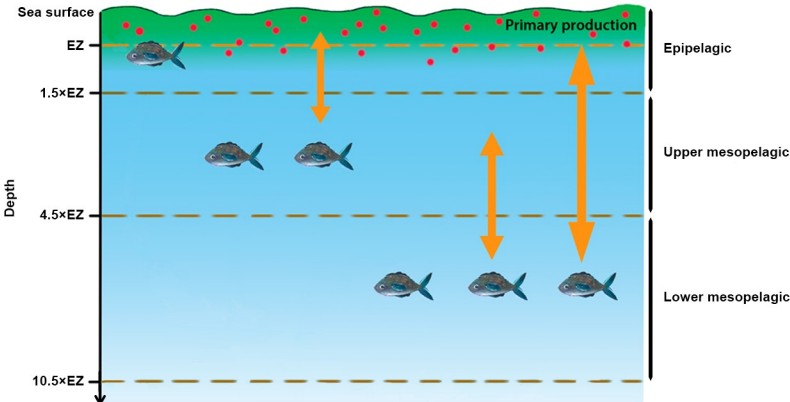

**Fig 2. Vertical repartition of the functional groups of micronekton in SEAPODYM in the water column.** The three depth layers, epipelagic, upper mesopelagic and lower mesopelagic layers are defined in SEAPODYM as multiples of EZ (depth of the euphotic zone set maximum at 1,000 m); fishes represent the micronekton functional groups; the red dots represent the zooplankton and the yellow arrows represent the daily vertical migrations of the functional groups between layers.

## Habitat-based density modelling

First, we modelled the distribution of beaked whales and sperm whales based on two habitat models: one with physiographic and oceanographic variables ('ENVIRONMENTAL model') and one with SEAPODYM variables ('SEAPODYM model'). We fitted GAMs [48, 49] with a Tweedie distribution to account for over-dispersion in the cetacean count data [50] with the 'mgcv' R-package [51]. GAMs extend Generalised Linear Models to allow for smooth nonlinear functions of predictor variables to be determined by data rather than by strict parametric relationships [48, 49]. The mean number of individuals per segment was linked to the additive predictors with a log-function with the dimension of the basis used to represent the smooth term fixed to a maximum of four to attenuate the scope for over-fitting [49]. An offset equal to segment length multiplied by twice the ESW or twice the 300 m-strip for JNCC-ESAS surveys was included ([52]; refer to Virgili et al. [27] for the ESW estimation; ESWs were calculated for each combination of platform—plane or boat, class of Beaufort sea-state and class of observation height). We removed combinations of variables with Pearson correlation coefficients higher than |0.7|and tested all models with combinations of one to four variables to avoid excessive complexity [53]. Environmental variables were poorly correlated with each other and moderately correlated with the SEAPODYM variables (S3.1 Fig in S3 Appendix). However, some SEAPODYM variables were correlated with each other, notably the biomass and production variables in the same layer. The Akaike information criterion (AIC, the lower the better; [54]) and Akaike weight ('qpcR' R-package; [55]) were used for model selection.

Then we built a model, for beaked whales and sperm whales, which combined environmental and SEAPODYM variables ('COMBINED model'). To limit the computational burden, mostly induced by the extent of the study area, we reduced the number of variables implemented in the variable selection procedure of the COMBINED model. Following Symonds & Moussalli [56], we determined the importance of each variable in the ENVIRONMENTAL and SEAPODYM models by summing the Akaike weights of the models in which the variable was selected and ranked all variables (S3.2 Fig in S3 Appendix). We then included in the selection procedure of the COMBINED models all variables whose percentages of Akaike weight were greater than 25%, as these were the most important variables in the ENVIRONMENTAL and SEAPODYM models. As for the ENVIRONMENTAL and SEAPODYM models, all models with combinations of one to four variables were tested and the best COMBINED model with the lowest AIC and the highest Akaike weight was selected. By design, we expected a better fit of the COMBINED model because we only used the most important variables selected in the first procedure.

Predictions of relative densities were then provided at 0.25˚ resolution over the entire study area. There were not enough data to fit a model by month or by season (the number of sightings in winter was too low) so we fitted ENVIRONMENTAL, SEAPODYM and COMBINED models to all data of beaked whales and sperm whales but we predicted relative densities for each month of the period and averaged values to obtain climatological prediction maps for the 1998–2015 period. These predictive maps provided the expected distribution of beaked and sperm whales according to static and monthly oceanographic conditions, or prey distributions or a combination of both. To provide uncertainty maps over the 1998–2015 period, averaged variances around the overall predictions were computed as the mean of the standard errors of the monthly predictions; high variances indicate high errors associated with density estimates. Here we only considered uncertainty associated with the densities predicted with the generalised additive model but we ignored other sources of uncertainty such as those associated with the estimation of the SEAPODYM variables.

To model distributions of beaked whales and sperm whales in the North Atlantic Ocean, we gathered survey data from a large region collected over a long period. The cumulative effort

was not homogeneously distributed and showed extensive geographical gaps. Consequently, predicting across the entire Atlantic basin would require extensive geographical extrapolation. Therefore, we conducted a gap analysis on environmental space coverage to identify areas where habitat models could produce reliable predictions outside survey blocks, *i.e.* geographical extrapolation, whilst remaining within the ranges of surveyed conditions for the combinations of covariates selected by the models, *i.e.* areas of environmental interpolation [57–59], as in Virgili et al. [27]. We estimated the convex hull defined by the environmental data used to fit habitat models *i.e.* the smallest convex envelop that contains all the points in environmental space. We assessed whether a prediction from a set of environmental covariates fell inside or outside this convex hull with the 'WhatIf' R-package [60–62]. A combination of climatological predictor values that fall inside the convex hull corresponds to an interpolation. We produced maps delineating the extent of interpolation areas and overlaid them on the density prediction maps to highlight areas with greater reliability and showing that no prediction was made outside interpolation areas.

Finally, model fit was assessed with the percentage of explained deviance [49, 63], the Root Mean Squared Error (RMSE) which measures prediction errors and model accuracy (the lower, the better; [64]; 'qpcR' R-package [55]); and a visual inspection of predicted and observed distributions [65]. This allowed a comparison of the three models in order to assess which combination of variables better explained the distribution of beaked and sperm whales.

## Results

### Beaked whales

The COMBINED model was identical to the ENVIRONMENTAL model as no SEAPODYM variable was selected (Fig 3A and S3.1 Table in S3 Appendix). The percentage of explained deviance varied between the models (Fig 3 and S3.1 Table in S3 Appendix). The explained deviance of the ENVIRONMENTAL/COMBINED model was higher than the explained deviance of the SEAPODYM model (39.8% vs 27.5%). The AIC and the RMSE were the lowest for the ENVIRONMENTAL/COMBINED model (ΔAIC = 335 for the SEAPOYM model and 0.49 vs 0.54 for the RMSEs; Fig 3).

In the best ENVIRONMENTAL/COMBINED model (Fig 3 and S3.1 Table in S3 Appendix), the highest beaked whale relative densities were found at depths *ca.* 2,000 m, spatial gradients of SST *ca.* 2˚C, slope *ca.* 1.5˚ and SST higher than 15˚C (Fig 3A). The highest relative densities were predicted on the western side of the North Atlantic Ocean near the Gulf Stream, along continental slopes and along the Mid-Atlantic Ridge (Fig 4A).

In the best SEAPODYM model (Fig 3 and S3.1 Table in S3 Appendix), the highest beaked whale densities were found at biomass of lower mesopelagic organisms *ca.* 1.5 g.m$^{-2}$, biomass of migrant upper mesopelagic organisms lower than 2 g.m$^{-2}$ per unit surface area, low production of migrant lower mesopelagic organisms (lower than 0.02 g.m$^{-2}$.day$^{-1}$) and deep euphotic depth (*ca.* 65 m; Fig 3B). This resulted in a fairly homogenous prediction of beaked whales in oceanic waters beyond the continental slope (Fig 4B).

Interpolation areas varied between the two models with the highest percentage of interpolation observed for the ENVIRONMENTAL/COMBINED model (85%) and the lowest for the SEAPODYM model (74%; Fig 4). In all cases, variances associated with the predictions were small (S4 Fig in S4 Appendix).

### Sperm whale

The percentage of explained deviance varied between the three models (Fig 3 and S3.2 Table in S3 Appendix). The explained deviance of the ENVIRONMENTAL model (31.1%) was close to

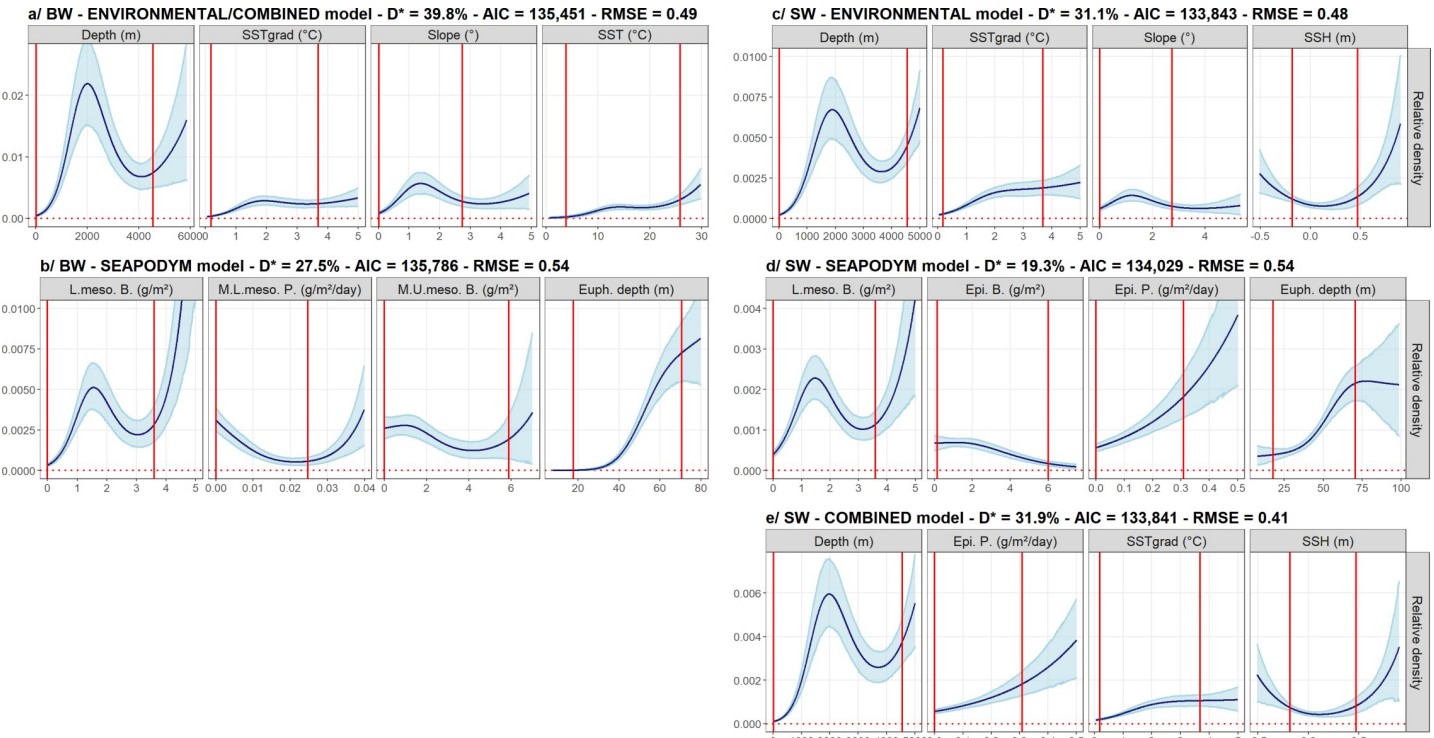

**Fig 3. The functional relationships between beaked (BW) and sperm (SW) whales and the selected variables of the ENVIRONMENTAL (a, c), SEAPODYM (b, d) and COMBINED (a, e) models.** The ENVIRONMENTAL and COMBINED models were identical for beaked whales thus represented only once. Solid lines represent the estimated smooth functions and the blue shaded regions show the approximate 95% confidence intervals. The relative density of individuals (individuals per 750 km²) is shown on the y-axis, where a zero indicates no effect of the covariate. Red vertical lines indicate the 2.5th and 97.5th percentile intervals corresponding to the core data. D*: explained deviance; AIC: Akaike criterion; RMSE: Root Mean Squared Error; SSTgrad: gradients of Sea Surface Temperature; L.meso. B.: lower mesopelagic biomass; M.L.meso.P.: migrant lower mesopelagic production; M.U.meso.B.: migrant upper mesopelagic biomass; Euph. depth: euphotic depth; SSH: sea surface height; Epi. B.: epipelagic biomass; Epi. P.: epipelagic production.

that of the COMBINED model (31.9%) and higher than that of the SEAPODYM model (19.3%). The AICs were identical in the ENVIRONMENTAL and COMBINED models and lower than the AIC of the SEAPODYM model (ΔAIC = 188 and 186 with the SEAPOYM model; Fig 3). The RMSE was the lowest for the COMBINED model (0.41) compared to the RMSEs of the ENVIRONMENTAL (0.48) and the SEAPODYM (0.54) models.

In the best ENVIRONMENTAL model (Fig 3 and S3.2 Table in S3 Appendix), the highest sperm whale densities were found at depths *ca.* 2,000 m, SSH higher than |0.1 m|, spatial gradients of SST higher than 2˚C and slope *ca.* 1˚ (Fig 3C). The highest densities were predicted on the western side of the North Atlantic Ocean, along continental slopes and along the Mid-Atlantic Ridge (Fig 4C).

In the best SEAPODYM model (Fig 3 and S3.2 Table in S3 Appendix), the highest densities were found at biomass of lower mesopelagic organisms *ca.* 1.5 g.m$^{-2}$, low biomass of epipelagic organisms (lower than 2 g.m$^{-2}$), production of epipelagic organisms higher than 0.2 g.m$^{-2}$.day$^{-1}$ and deep euphotic depth higher than 60 m (Fig 3D). This resulted in fairly homogenous predicted densities in oceanic waters (Fig 4D).

In the best COMBINED model (Fig 3 and S3.2 Table in S3 Appendix), the highest densities were found at depths *ca.* 2,000 m, production of epipelagic organisms higher than 0.2 g.m$^{-2}$.day$^{-1}$, gradients of SST higher than 2˚C and SSH higher than |0.1 m| (Fig 3E). This resulted in similar patterns as for the ENVIRONMENTAL model but with more homogenous densities (Fig 4E).

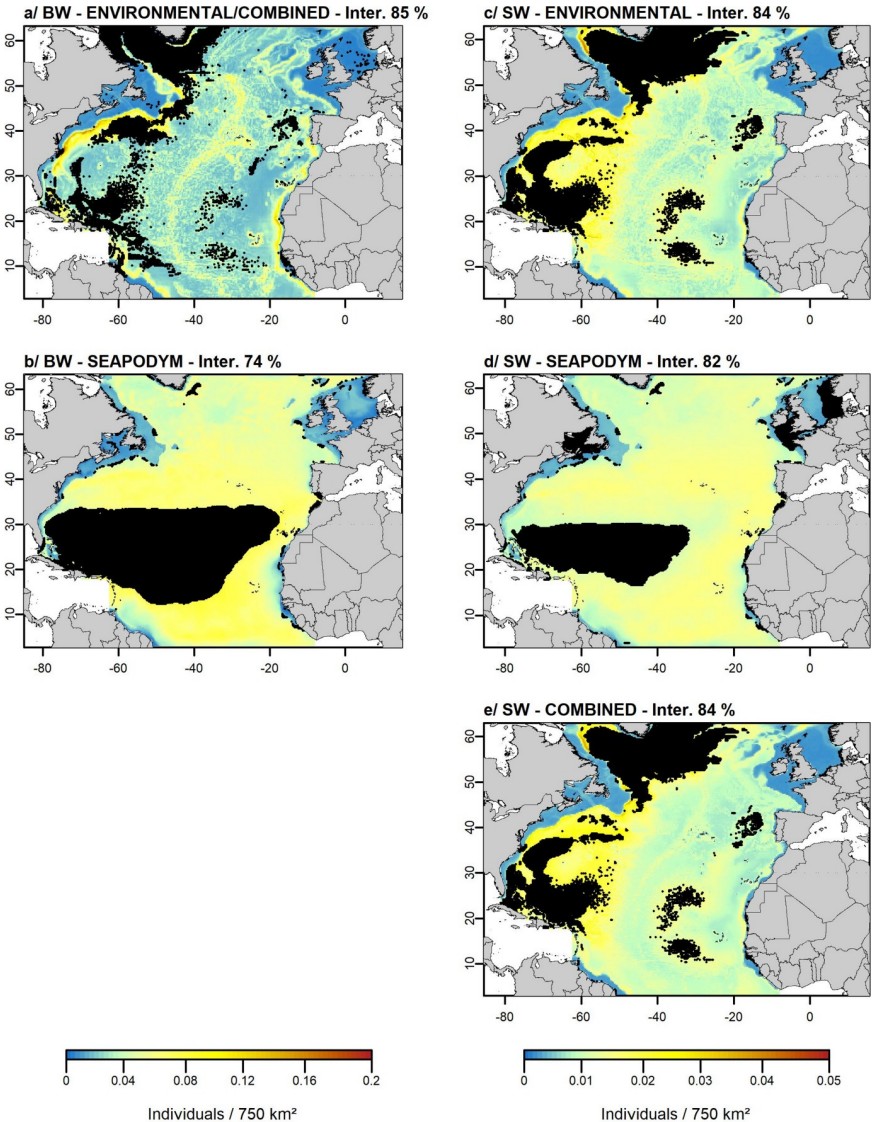

**Fig 4. The predicted relative densities of beaked (BW) and sperm (SW) whales obtained from the ENVIRONMENTAL (a, c), SEAPODYM (b, d) and COMBINED (a, e) models.** A square root scale has been used to visualise the predictions but the values in the legend represent the untransformed predicted values. The ENVIRONMENTAL and COMBINED models were identical for beaked whales thus represented only once. Black areas on prediction maps represent zones where we did not extrapolate the predictions. Percentages represent the proportion of the study area defined as interpolation (Inter.) with the gap analysis. Base map from https://www.gebco.net/.

Interpolation areas varied between the three models and were less extended for the SEAPODYM model (82%) compared to the ENVIRONMENTAL and COMBINED models (84%; Fig 4). In all cases, variances associated with the predictions were small (S4 Appendix).

## Discussion

Indirect proxies of prey distribution such as SST or SSH are routinely used in habitat models to describe marine top predator distributions because prey data are generally unavailable. The relationships estimated between these proxies and predator occurrence or abundance are

therefore indirect. The integration of higher trophic levels in habitat models could improve the model performances and the ecological explanation of the relationships, as shown in Lambert et al. [21] or Putra & Mustika [66]. In this study, we incorporated potentially more direct variables into habitat-based density models by using simulated distributions of functional groups of prey. The models were fitted to sightings of deep-diving cetaceans (beaked and sperm whales) because their habitat use is still poorly known and they face important anthropogenic threats including activities producing high intensity noise (*e.g.*, military sonars or seismic guns [29, 67]), which makes them species of major interest. For species that spend most of their time at depth, like deep-divers, information on potential prey in deep layers could improve current habitat models because they are mostly based on surface variables and not on variables that characterise deep layers [27, 32, 33, 68].

## Methodological considerations

It may seem inappropriate to use prey simulations at depth to explain the distribution of surface sightings, it would be more appropriate to use tracking data [22]. However, the accepted underlying assumption (probably stronger than it is in reality) is that animals observed at the surface are present because they are mostly sensitive to the prey abundance [11, 15–18] and therefore a high prey biomass at depth could explain the presence of animals at the surface. Very few studies to date have used subsurface or depth variables to explain the distribution of predators. Brodie et al. [69] found the inclusion of two subsurface dynamic variables (bulk buoyancy frequency and isothermal layer depth) in species distribution models to better describe the habitats of four pelagic species in the California Current System and to increase model explanatory power and predictive performance for most species. A more systematic use of depth variables could improve the tools available for the planning of human activities, especially for species that would be closely linked to processes at depth.

The combination of sighting data from different ecosystems (*e.g.* North-East and North-West Atlantic) may mask inter-regional differences in the relationships between cetacean densities and the environmental predictors, as shown by Mannocci et al. [70] and may mask the influence of prey data on species distribution. The objective here, was to compare models using environmental variables, prey distribution variables, and a combination of the two types of variables to determine the extent to which the models could be improved and not necessarily explain precisely the mechanisms influencing the distribution of beaked and sperm whales, which would be more consistent at the scale of a smaller region. We built basin-wide models and did not investigate region-specific models (*sensu* [70]) in order to leverage large sample sizes for investigating the explanatory power of SEAPODYM variables for habitat use of deep-divers. We acknowledge that pooling regions across an oceanic basin may introduce bias, but this study is a first investigation, and appreciate region-specific models would represent an obvious improvement (once enough data are available at the regional scale). In a global conservation context, it is also important to study the broad scale cetacean distribution to obtain a more global prediction of this distribution.

One of the problems faced when studying the distribution of deep-diving cetaceans through habitat modelling is the lack of sighting data, it is necessary to assemble datasets from different surveys to obtain sufficient data to fit habitat models [27, 32, 68]. Sampling is therefore non-uniform, both spatially and temporally, with some areas, years or seasons being much more represented. For example, some years were only sampled in the northwest Atlantic area. Therefore, we chose to combine all the data to ensure we had enough data without considering a temporal effect in the models. However, temporality was taken into account when associating environmental conditions with effort segments, so when there was a sighting, it was associated with

environmental conditions averaged over the week for the SEAPODYM variables and over the month for the environmental variables before the sighting, to take into account potential time-lags [11–13]. When deep-diver data will be large enough to allow a more uniform temporal sampling, the temporal aspect will be taken into account in the models.

We took care in using appropriate statistical tools for modelling the habitat of species with few sightings [71]. Virgili et al. [71] showed that GAMs with a Tweedie distribution generated reliable habitat modelling predictions for rarely sighted marine predators. We constrained the number of variables in the models to a maximum of 4 (ensuring that they were all significant) to avoid excessive complexity of models and difficulty in their interpretation [53]. Although this could limit the purely predictive performance of our models, we were confident in the procedure parsimony for explanatory purposes. Indeed, the explained deviances obtained (between 20 and 40%) were high compared to other studies on cetaceans [63, 68], suggestive of a good fit to the data, and for each model, only four or five variables were actually important, the others representing a very small percentage of the Akaike weights (S3.2 Fig in S3 Appendix). As a result, four variable models were the best compromise between avoidance of overfitting, reduction of complexity and maximisation of interpretability.

## Ability of SEAPODYM and environmental variables to model deep-diver distributions

The model comparison showed that for beaked whales, the explained deviance was higher and the RMSE was lower in the ENVIRONMENTAL model than in the SEAPODYM model, so environmental variables seemed to better explain their distribution than SEAPODYM variables; this could explain why no SEAPODYM variable was selected in the COMBINED model (most important variables for the beaked whale COMBINED model were only environmental variables; S3.2 Fig in S3 Appendix). As for Torres et al. [72], it is possible that the integration of prey data did not improve the explanatory power of the models because the prey sampling was not appropriate or there was a mismatch between the distribution of prey and predators because they respond to oceanographic processes at different spatial and temporal scales. The link between predators and the environment may therefore be simpler to model than the link between predators and prey [72] or the spatial resolution of the SEAPODYM variables (0.25˚) was quite coarse to depict specific associations between an animal and its environment. In contrast, for sperm whales, the ENVIRONMENTAL and COMBINED models were comparable but the explained deviance and the RMSE were slightly better for the COMBINED model so SEAPODYM variables seemed to provide additional information to that provided by the environmental variables alone. The production of epipelagic organisms appeared to be one of the five most important variables in the sperm whale COMBINED model (S3.2 Fig in S3 Appendix). These results agreed with those of Hazen et al. [16] who showed that the performance of a model which included static (depth), oceanographic (salinity and temperature) and biological (deep scattering layer and number of available prey) variables was higher than that of the model which only included prey data. One hypothesis, suggested by Hazen et al. [16], would be that beaked whales do not necessarily track a particular type of prey but rather large-scale physiographic structures while sperm whales would be more pelagic predators, so the prey simulated by SEAPDOYM would correspond more closely to the prey targeted by sperm whales than by beaked whales. Another hypothesis could be a temporal mismatch between SEAPODYM output and the growth rate of the actual prey of these species as discussed by Chambault et al. [73].

The functional groups of prey that were simulated from SEAPODYM may not directly reflect the prey targeted by deep-divers. The three biological layers of SEAPODYM are defined

between 0 and *ca.* 1,000 m with organisms ranging from 2 to 20 cm [44, 45], but deep-divers can feed beyond 1,000 m on prey bigger than 20 cm, mostly beaked whales [31, 74, 75]. From stomach contents, Spitz et al. [31] showed that the diet of Cuvier's beaked whales consisted largely of cephalopods bigger than 20 cm, *Teuthowenia megalops* (21,2 ± 3,3 cm) and *Galiteuthis armata* (24,2 ± 2,4 cm) and the diet of sperm whales was mostly composed of cephalopods close to 20 cm, *Histioteuthis bonnellii* (17,8 ± 3,7 cm) in the north-east Atlantic Bay of Biscay. Thus, the functional groups of prey simulated in SEAPODYM may represent only a part of the prey targeted by deep-divers and the probable link could be mediated by deep-diver prey instead. The prey simulated in SEAPODYM may correspond more closely to the prey of the prey targeted by deep-divers. A similar phenomenon was observed in the study of Putra & Mustika [66] which showed that the zooplankton biomass was a strong predictor of the distribution of manta rays (*Mobula birostris*) because they feed directly on zooplankton whereas it was not a strong predictor for dolphins (*Stenella* spp.) because they are second- and third-level consumers. Nevertheless, the study shed light on dolphin foraging habits. It is likely the same situation in our study, the prey targeted by deep-divers may be larger than those simulated in SEAPODYM [31], therefore the simulated trophic levels may be lower than those actually targeted by deep-divers. This may explain why epipelagic organisms were retained in the sperm whale models, many bathypelagic squids and fishes are vertically migrating predators feeding at night in the epipelagic layer. In addition, the functional groups simulated by SEAPODYM encompass multiple organisms (fishes, cephalopods, crustaceans; [44, 45]), so deep-diver prey, which are mainly cephalopods, might be poorly represented in the functional groups in terms of biomass and production, even in upper and lower mesopelagic layers, explaining why they were not selected in the COMBINED models.

Although it would be more consistent to fit models at a regional scale, we highlighted interesting relationships with the environment and/or the simulated prey at a large scale. Relationships obtained in the ENVIRONMENTAL models suggested a preference for habitats associated with topographic features and thermal fronts for both species, as previously shown [27, 32, 33]. Comparatively, SEAPODYM variables reveal areas of prey concentration whereas environmental variables are only indirect proxies of the prey distribution. Putra & Mustika [66] suggested that chlorophyll *a* concentrations are not spatially correlated to high zooplankton biomasses because of the time lag between phytoplankton development and meso-zooplankton growth, therefore dolphin (*Stenella* spp.) densities would be higher where zooplankton biomasses would be higher but where chlorophyll *a* concentrations would be lower. SEAPODYM variables could thus give a better idea of the foraging areas used by the species [19, 23, 45]. We found that beaked whales were closely related to quite high biomass of lower mesopelagic organisms, low biomass of migrant upper mesopelagic organisms, low production of migrant lower mesopelagic organisms and deep euphotic depth. This would suggest a preference for habitats where the euphotic zone is extended and with large prey of the lower mesopelagic layer but not a preference for large organisms of the upper mesopelagic layer, which could partly correspond to the prey targeted by beaked whales [31]. Sperm whales were closely related to quite high biomass of lower mesopelagic organisms, low biomass of epipelagic organisms, high production of epipelagic organisms and deep euphotic depth. As for beaked whales, this would suggest a preference for habitats where the euphotic zone is extended, with large prey of the lower mesopelagic layer but also abundant small epipelagic prey *i.e.* a productive habitat at the surface which allow the development of large prey at depth. This is partly consistent with the diet of sperm whales which seems more flexible than the diet of beaked whales [31]. The positive relationship with epipelagic production and the negative relationship with epipelagic biomass would indicate habitats with abundant small epipelagic prey and not large epipelagic prey, which would be consistent with the previously mentioned

hypothesis that this would be the prey targeted by sperm whale prey. The variables selected in the COMBINED models (or ENVIRONMENTAL model for beaked whale) would suggest that beaked whales were more associated with physiographic features around 2,000 m deep so they might forage on organisms living close to the seabed that may be poorly modelled by SEAPODYM. By contrast, sperm whales would be less constrained by the presence of the slope, and would therefore prey on organisms that are truly pelagic [76], and therefore possibly better modelled by SEAPODYM. The addition of other depth variables such as the bottom temperature could help clarify the relationship with depth [77]. Other variables could also be considered in the models such as SSH fronts, characteristics of the eddies (*e.g.* radius, type), however the objective here was to compare SEAPDOYM variables to environmental surface variables commonly used and known to influence the deep-diver distribution [27, 32, 33, 68]. It would be particularly interesting to characterise eddies (*e.g.* cyclonic or anticyclonic) to assess whether deep-divers preferentially select a certain type of eddy as it would appear that prey aggregating at eddies are different depending on their rotation cycles [78].

The euphotic depth, which is the maximum depth of the light zone suitable for phytoplankton photosynthesis, was selected for both beaked whales and sperm whales in the SEAPODYM models with increasing relationships. The euphotic depth was a variable available in SEAPODYM but the mixed layer depth would probably be more informative for deep-divers because it provides information on the depth of the thermocline which is frequently used by diving animals [77], it would therefore be interesting to explore the differences in the relationship between these two types of variables for deep-divers.

For both beaked and sperm whales, depth was selected in the models. There is a great interest in including depth in the models because it is a well-measured variable, *i.e.* with low associated measurement error, unlike remote sensed or modelled variables which may help improving the explanatory power of the models. Pendleton et al. [79] observed similar results for bowhead whales (*Balaena mysticetus*). They compared a model using only environmental and biological variables such as sea ice thickness, sea temperature, diatoms, flagellates, copepods and zooplankton from the Biology Ice Ocean Modeling and Assimilation System (BIOMAS) to a model using these variables plus bathymetry and showed that the best model included bathymetry and BIOMAS variables. The importance of bathymetry suggests there is probably a close relationship between bathymetry and the biophysical processes that determine the distribution of these cetaceans.

Overall, no beaked whales and sperm whales were predicted on the continental shelf because no sightings were recorded, which confirms the non-suitability of this habitat for these species [32, 33]. The bathymetry greatly influenced the beaked and sperm whale distributions in the ENVIRONMENTAL and COMBINED models, with the highest densities predicted near slope discontinuities where the bottom is steep and where prey aggregate [80, 81]. A west-east density gradient was also observed for both species, which is consistent with the studies of Roberts et al. [32] and Rogan et al. [33]. Sperm whales, appeared to be less linked to these structures than beaked whales and their distribution was slightly more homogeneous. With SEAPODYM variables, predicted densities were lower and prediction maps were smoother (a common scale was applied to facilitate the comparison). Beaked and sperm whales seemed homogeneously distributed in oceanic waters, which was not very consistent with the sightings and highlighted the poorer performance of the models (higher RMSEs). As previously showed in Lambert et al. [21], these smoother predictions may be due to the large resolution of the variables. The spatial resolution of the SEAPODYM variables is currently set to 0.25˚ but is planned to be refined to 0.08˚ and a finer spatial resolution may reduce this smooth effect. However, by adding environmental variables, particularly depth, in the sperm whale COMBINED model, the model performance was increased (higher explained deviance and

lower RMSE) and the prediction was more spatially detailed with highest densities predicted in the western Atlantic.

We performed a gap analysis to highlight geographical extrapolation areas in which environmental and SEAPODYM variables were out ranges of surveyed conditions. The centre of the study area, near the Mid-Atlantic Ridge, was not surveyed but variable conditions remained within the ranges of surveyed conditions, allowing to predict distribution at a large scale. The gap analysis revealed large gaps in environmental space coverage across the study area, especially in the western part of the Atlantic Ocean for the ENVIRONMENTAL models and in tropical waters for the SEA-PODYM models. This suggests that sampling effort was not sufficient in deeper and steeper areas and more intensive sampling effort performed in these areas could help better describe the habitat used by deep-divers. The interpolation areas varied between the different models, from 74/82% for the SEAPODYM models to 84%/85% for the ENVIRONMENTAL and COMBINED models, which means that 18 and 26% of the area represented an environmental extrapolation with the SEAPODYM models, compared to only 16% with the COMBINED models for sperm whales and 15% in the ENVIRONMENTAL model for beaked whales. The uncertainties were thus greater with the SEAPODYM models and it appeared riskier to perform models at a large scale with SEA-PODYM variables only, than with environmental variables. For sperm whales, by combining both types of variables, this risk was slightly reduced compared to the SEAPODYM model and areas of concentration seemed more spatially detailed and better characterised with the COMBINED model, as revealed by a lower RMSE. To meet sperm whale conservation objectives, it seems more valuable to use models combining environmental variables and prey distribution data. Habitat models are regularly used to identify areas where species concentrate in order to set up protection areas [82–84]. They are also useful for predicting species distribution in non-sampled or poorly documented areas [85, 86], provided that areas of interpolation and extrapolation are considered [27, 57, 58]. Considering key habitats of species, particularly foraging habitats, in the design of marine protected areas thanks to the combination of environmental and prey variables (with further development to include prey of all types of predators) would represent an improvement in the tools available for species conservation, and consideration could be given to reducing the overlap between impacting human activities and these key habitats [87, 88].

## Conclusion

Prey distribution data are not widely available to model the distribution of marine top predators. However, there are models, still not widely used, such as SEAPODYM, which simulate the distribution of functional groups of prey. SEAPODYM did not seem to model accurately the prey of deep-diving cetaceans assuming the later are sighted at the surface of their foraging grounds. By combining SEAPODYM and environmental variables the increase model performance for sperm whale was only marginal. For beaked whales, combining variables did not improve the model performance at all, no SEAPDOYM variable was selected in the best model, possibly because they feed on organisms more associated to seabed features and consequently less well predicted by SEAPODYM that focus on pelagic prey assemblages. This study was a first investigation and we mostly highlighted the importance of the physiographic variables to understand mechanisms that influence the distribution of deep-diving cetaceans. To improve the results, it would be interesting to refine spatial resolution of the variables, to fit region-specific models or to apply this approach to species whose functional groups of prey would be better represented in SEAPODYM such as baleen whales or small delphinids. A more systematic use of SEAPODYM could allow to better define the limits of its use and a development of the model that would simulate larger prey beyond 1,000 m would probably better characterise the prey of deep-diving cetaceans.

## Supporting information

**S1 Appendix. Details of surveys used in the analyses.** Total effort represents the total length of transects of each survey (without removing the transects with a Beaufort sea-state > 4). NE-ATL: Northeast Atlantic Ocean; NW-ATL: Northwest Atlantic Ocean.
(PDF)

**S2 Appendix. Average conditions of the static, oceanographic and SEAPODYM variables over the entire period (from 1998 to 2015).** Base map from https://www.gebco.net/.
(PDF)

**S3 Appendix. Correlations between environmental and SEAPODYM variables and model outputs for beaked and sperm whales.**
(PDF)

**S4 Appendix. Uncertainty maps representing the standard error associated with the predicted relative density of beaked (BW) and sperm (SW) whales.** Black areas represent extrapolation where we did not extrapolate the predictions. Base map from https://www.gebco.net/.
(TIF)

## Acknowledgments

We are grateful to the many observers who participated in the surveys and collected all the data but also the ships' captains, crews and pilots. We thank Phil Hammond and his team for providing SCANS and CODA survey data. THUNNUS survey was carried out thanks to the collaboration of the General Directorate of Fisheries and Maritime Affairs, Government of Galicia. We thank the *Direction Générale de l'Armement* (DGA), including Odile Gérard and Carole Nahum, for their support during this study. We warmly thank Charlotte Lambert for her help, support and comments. We thank Philippine Chambault and the two other anonymous referees, for their very helpful comments that led to a clearer and much improved manuscript.

## Author Contributions

**Conceptualization:** Auriane Virgili, Vincent Ridoux.

**Formal analysis:** Auriane Virgili, Laura Hedon, Matthieu Authier.

**Methodology:** Auriane Virgili, Laura Hedon, Matthieu Authier, Beatriz Calmettes, Patrick Lehodey, Pascal Monestiez.

**Resources:** Beatriz Calmettes, Diane Claridge, Tim Cole, Peter Corkeron, Ghislain Dorémus, Charlotte Dunn, Tim E. Dunn, Sophie Laran, Patrick Lehodey, Mark Lewis, Maite Louzao, Laura Mannocci, José Martínez-Cedeira, Debra Palka, Emeline Pettex, Jason J. Roberts, Leire Ruiz, Camilo Saavedra, M. Begoña Santos, Olivier Van Canneyt, José Antonio Vázquez Bonales.

**Supervision:** Pascal Monestiez, Vincent Ridoux.

**Validation:** Vincent Ridoux.

**Writing – original draft:** Auriane Virgili.

**Writing – review & editing:** Auriane Virgili, Laura Hedon, Matthieu Authier, Beatriz Calmettes, Diane Claridge, Tim Cole, Peter Corkeron, Ghislain Dorémus, Charlotte Dunn, Tim

E. Dunn, Sophie Laran, Patrick Lehodey, Mark Lewis, Maite Louzao, Laura Mannocci, José Martínez-Cedeira, Pascal Monestiez, Debra Palka, Emeline Pettex, Jason J. Roberts, Leire Ruiz, Camilo Saavedra, M. Begoña Santos, Olivier Van Canneyt, José Antonio Vázquez Bonales, Vincent Ridoux.

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
