## [Decision Letter · Decision Letter 0]

15 Mar 2021

PONE-D-21-03978

Towards a better characterisation of deep-diving whales’ distributions by using prey distribution model outputs

PLOS ONE

Dear Dr. Virgili,

Thank you for submitting your manuscript to PLOS ONE. After careful consideration, we feel that it has merit but does not fully meet PLOS ONE’s publication criteria as it currently stands. Therefore, we invite you to submit a revised version of the manuscript that addresses the points raised during the review process.

We look forward to receiving your revised manuscript.

Kind regards,

Vitor Hugo Rodrigues Paiva, Ph.D.

Academic Editor

PLOS ONE

Journal Requirements:

'The authors have declared that no competing interests exist.'

We note that one or more of the authors are employed by a commercial company: ADERA

4. We note that Appendix S2, S4 and Figures 1 and 4 in your submission contain map/satellite images which may be copyrighted.

a. You may seek permission from the original copyright holder of Appendix S2, S4 and Figures 1 and 4 to publish the content specifically under the CC BY 4.0 license. 

Reviewers' comments:

Reviewer's Responses to Questions

**Comments to the Author**

1. Is the manuscript technically sound, and do the data support the conclusions?

Reviewer #1: No

Reviewer #2: Partly

Reviewer #3: Partly

2. Has the statistical analysis been performed appropriately and rigorously? 

Reviewer #1: Yes

Reviewer #2: Yes

Reviewer #3: No

3. Have the authors made all data underlying the findings in their manuscript fully available?

Reviewer #1: Yes

Reviewer #2: Yes

Reviewer #3: Yes

4. Is the manuscript presented in an intelligible fashion and written in standard English?

Reviewer #1: Yes

Reviewer #2: Yes

Reviewer #3: Yes

5. Review Comments to the Author

Reviewer #1: The article entitled "Towards a better characterisation 1 of deep-diving whales’ distributions by using prey distribution model outputs" aimed at comparing 2 habitat modelling approaches to predict the distribution of deep divers using surface environmental variables vs. low and mid-trophic levels outputs. The authors showed that including SEAPODYM covariates does not improve the model performance and conversly lead to drastic different predictions, raising serious concern regarding the reliability of the model output and the algorithm used (classical additive model). Despite the efforts of the authors to provide a clear methodology, the results lack of clarity (numerous repetitions, the results are not sufficiently discussed in regards to other studies) and the bias / limitations of the modelling procedure are not well discussed. I suggested several improvements, including testing other algorithms from machine learning and adding other covariates into the model as well as additional litterature. The authors have a dataset will a great potential, and I'm convinced once the methodology will be made more robust and the results clearer, this article will be of great contribution to the habitat modelling and marine mammals communities!

Reviewer #2: Virgili et al. have identified an important gap in our understanding of the link between potential habitat for deep-diving marine predators and the distribution of those predators- namely the influence of prey on the distribution of predators as an intermediate explanatory step between the environment and predators. The most important finding of this work is that the SEAPODYM model of pelagic fish distribution does not explain the distribution of deep-diving predators better than environmental data (except for some weak evidence for sperm whales).

While the science in the paper is basically sound (though see comment below regarding model derivation), the framing needs a bit of work before it should be published. That is, what is the driving hypothesis of this paper? There seem to be two possible frameworks:

1) The current hypothesis is that SEAPODYM variables better explain the distribution of beaked and sperm whales. This was found not to be the case (that is, this is a negative result). If this framework is preferred, it should be stated in the abstract, introduction and conclusion very clearly that SEAPODYM was ruled out as a good framework for describing the distribution of these deep divers

2) Given that, as stated in the discussion lines 396-424, SEAPODYM variables attempt to predict pelagic fish concentrations shallower than 1000m (that is, not the preferred prey of the study predators), a better test of SEAPODYM would be to choose a different predator to examine. Perhaps tuna or some other publicly available dataset (see e.g. https://portal.atn.ioos.us/) could be used to test the model more appropriately?

Besides framing, the other major issue with the manuscript in its current form is that it does not appear that the SEAPODYM model has fundamentally different information than the ENVIRONMENTAL model, thus it is not surprising that it does not improve the explanatory power of the environmental model. That is, in the paragraph on line 160, the SEAPODYM model appears to derive from SST, bathymetric and surface current data, which is the same information that parameterizes the ENVIRONMENTAL model. Is not one then just a transformation of the other? What information does the SEAPODYM model have the that the environmental model does not. There may be something in how data was integrated over time, but it was not clear from the current manuscript how these models are substantially different. In the revision, please clarify how these models provide different information, given that they do not actually include information about prey in the SEAPODYM model.

Minor comments:

Abstract: please include a stronger summary in the abstract describing the main results (that SEAPODYM is not a good predictor of deep-diving cetacean habitat).

Line 75- also see croll et al.

Croll, D. A., Marinovic, B., Benson, S., Chavez, F. P., Black, N., Ternullo, R., & Tershy, B. R. (2005). From wind to whales: trophic links in a coastal upwelling system. Marine Ecology Progress Series, 289, 117-130.

Line 98- not clear what the difference between seapodym and seapodym model are?

Line 166- Why a 29 day sampling window? It seems like this may be appropriate for some variables but for others it would smooth over important information. A sensitivity analysis of choice of sampling window should be considered

General point of clarification- the spatial aspect of the modeling seems clear, but it would be helpful to have more explicit discussions of how the temporal aspect of the modeling was done, particularly for predator abundances that shift over time. Was there repeat sampling done or is predator abundance from a single survey? The “gapping” described appears to refer mostly to missing spatial data.

Reviewer #3: Overview and general comments

The main objective of this paper is to compare habitat models fit to different types of variables to for two deep diving cetaceans: sperm whales and beaked whales. The authors used generalized additive models (GAMs) to compare habitat models fit to 1) environmental variables, 2) simulated prey data from Spatial Ecosystem And POpulation DYnamics Model (SEAPODYM), and 3) combinations of the two variable types. The purpose of this study is to investigate if more direct measures of prey improve the fit of habitat models, when compared to prey proxies such as surface chlorophyll. The authors predict that more proximal variables, like prey biomass (simulated by SEAPODYM) will improve habitat models by reducing the time lag between phytoplankton biomass and the effect on higher trophic levels. The authors have presented and interesting and relevant study, however there are some issues with the methodology that must be addressed prior to publication. Major and minor concerns are detailed below.

Major Concerns

1. My first concern is within the ‘COMBINED’ habitat models for both species. The authors stated that they limited the computational burden within the combined model by only included a reduced number of variables; specifically, by only including variables selected within the final “ENVIRONMENTAL” and “SEAPODYM” models. This strikes me as a red flag and limitation. All relevant variables should be included in the ‘COMBINED’ models. If a variable is not significant in the “ENVIRONMENTAL” model, for example, this does not mean it will automatically not be significant in the “COMBINED” model. Specifically, an interaction between an environmental variable and a SEAPODYM variable could be significant in the “COMBINED” model, and this of course would not show up in the individual “ENVIRONMENTAL” or “SEAPODYM” models.

2. The authors do not reference/mention spatial auto-correlation or how it was accounted for within the models (e.g. include correlation structure in the model, bin the data at resolutions that are broader than the correlation).

3. Explained deviance and AIC are great for model comparison and selection, but are insufficient in terms of model evaluation. AIC and explained deviance are what you use to choose your model, but doesn’t tell you anything about model performance. I would like to see an evaluation metric such as area under the curve (AUC) or true skill statistic (TSS). AUC indicates model performance, provide aggregate measures of performance across all possible classification thresholds, and allow performance of models to be compared. TSS measure the discriminatory ability of the SDMs (e.g. Becker et al. 2020).

4. A lesser concern is with the observations. As I am not a deep-diving whale expert, I am having trouble reconciling the use of surface observation for a study that focuses on fitting habitat models to sub-surface prey simulations. I am not contending that the use of surface observation is inappropriate, however I do think the methods would benefit from more explanation. For example, do we know how surface observation relates to deep dives for prey? Also including a couple references for studies that have used this type of observation for deep diving species’ habitat models previously would be useful.

Minor comments

Line 57-59: “cetacean distributions are barely known” is an overgeneralization. If you are referring to deep-diving cetaceans, this needs to be made more explicit. Distributions of some cetaceans are actually quite well known and well modeled (e.g. Abrahms et al. 2019 – blue whales).

Lines 291-294; 307, etc.: The way the authors refer to the “COMBINED” model for the beaked whale is confusing. The authors should state that the “COMBINED” model is identical to the “ENVIRONMENTAL” model at the beginning of the results section referencing beaked whales. Stating the comparison of AIC scores prior to making this point is confusing and makes it seem like these are different models (i.e. contain different variables). Similarly, within the Figure 3, it seems unnecessary to show identical maps for 3a and 3c. Instead, the authors should simply indicate that the “COMBINED” model is identical, and not in fact a separate model with separate results.

Line 333-334, etc.: Remove line “All selected variables were poorly correlated” lines from the results section. This is inherent based on the methodology. As such, you would only need to report if variables were slightly correlated, e.g.

Line 418-424: I did find myself wondering the need to combine observation across such a large region. The authors offer an explanation in lines 418-424, but I don’t think this explanation is sufficient. Why not focus this study on one of the regions referenced? Would the authors expect prey data to be more relevant and improve fit at regionally specific scales? Is it possible that such a broad spatial area masks the impacts of prey data?

Lines 483-484: Generally, in this part of the discussion, the authors continue to refer to the “COMBINED” models (plural) limiting the uncertainty within the interpolation. However, the authors previously state that the “COMBINED” model is identical to the “ENVIRONMENTAL” model for beaked whales, meaning that these conclusions would not be true for the beaked whales. The authors need to specifically reference the sperm whale example when making these statements/conclusions.

References:

Abrahms, Briana, et al. "Dynamic ensemble models to predict distributions and anthropogenic risk exposure for highly mobile species." Diversity and Distributions 25.8 (2019): 1182-1193.

Becker, Elizabeth A., et al. "Performance evaluation of cetacean species distribution models developed using generalized additive models and boosted regression trees." Ecology and evolution 10.12 (2020): 5759-5784.

6. PLOS authors have the option to publish the peer review history of their article (what does this mean?). If published, this will include your full peer review and any attached files.

Reviewer #1: **Yes: **Philippine Chambault

Reviewer #2: No

Reviewer #3: No

---

## [Author Response · Author response to Decision Letter 0]

19 May 2021

Dear Editor,

Thank you for considering our article entitled “Towards a better characterisation of deep-diving whales’ distributions by using prey distribution model outputs”, for publication in the journal PLOS ONE. We have taken into account all comments from the editor and the three reviewers regarding the article. 

We thank the reviewers for their careful reading of our manuscript and for pointing out elements in the abstract, methods and discussion that needed to be more developed. We did not change the models used in our study, as generalized additive models have repeatedly shown their effectiveness in modelling the distribution of cetaceans (e.g. Roberts et al., 2016; Becker et al., 2017; Cañadas et al., 2018; Virgili et al., 2019) but we have completed the methodology to make it more robust, we have changed the temporal resolution of the SEAPODYM variables so that the link between species distribution and these variables would be more direct, we have modified the process of variable selection for the COMBINED models, and we have added a new parameter for estimating the model performance. Many discussion points have been completed and added in order to meet the expectations of the reviewers. 

We have fixed all typos and grammatical mistakes during the revision and ensured that the manuscript meets the PLOS ONE’s style requirements. 

We hope that the revised manuscript addresses the different concerns raised by the editor and the reviewers, whom we thank for their careful reading and review of this manuscript. We feel the revised manuscript has improved as a result and hope it will be deemed acceptable for consideration of publication in PLOS ONE.

Sincerely,

On behalf of all co-authors,

Auriane Virgili

Concerning specific Editor and reviewer’s comments:

Editor’s comments: 

ANSWER: We have checked all the requirements provided on the website and modified the manuscript to meet these requirements.

'The authors have declared that no competing interests exist.'

We note that one or more of the authors are employed by a commercial company: ADERA

Please know it is PLOS ONE policy for corresponding authors to declare, on behalf of all authors, all potential competing interests for the purposes of transparency. PLOS defines a competing interest as anything that interferes with, or could reasonably be perceived as interfering with, the full and objective presentation, peer review, editorial decision-making, or publication of research or non-research articles submitted to one of the journals. Competing interests can be financial or non-financial, professional, or personal. Competing interests can arise in relationship to an organization or another person. Please follow this link to our website for more details on competing interests: http://journals.plos.org/plosone/s/competing-interests.

ANSWER: 

Funding statement

AV’s doctoral research grant was funded by the Direction Générale de l'Armement (DGA). 

ML was funded by a Ramón y Cajal (RYC-2012-09897) postdoctoral contract of the Spanish Ministry of Economy, Industry and Competitiveness. This study is a contribution to the CHALLENGES (CTM2013-47032-R) project of the Spanish Ministry of Economy, Industry and Competitiveness.

MA and EP are affiliated to ADERA a private business dedicated to providing support for the management of research project in public institutions in Region Nouvelle Aquitaine. In this study, ADERA solely provided support in the form of salaries for authors MA and EP, but did not have any additional role in the study design, data collection and analysis, decision to publish, or preparation of the manuscript. The specific roles of these authors are articulated in the ‘author contributions’ section.

Competing interest statement

MA and EP are employed by the commercial company ADERA which did not play any role in this study beyond that of employer. This does not alter our adherence to PLOS ONE policies on sharing data and materials.

ANSWER: We indicated the data from this study are available upon request because each of the co-authors of the manuscript owns part of the dataset we used and a request has been made to each of them to have access to the data so we cannot provide the dataset without their agreement. All sighting and effort data used in this study are available in the OBIS SEAMAP database: http://seamap.env.duke.edu/. Authors had no special access to the underlying data and all interested researchers can obtain the data from http://seamap.env.duke.edu in the same way that the authors obtained data by contacting the data providers via the OBIS SEAMAP website. 

Email address of organisations/data providers: 

AMBAR: ambarelkartea@gmail.com

AZTI: mlouzao@azti.es

BMMRO: dclaridge@bahamaswhales.org

CEMMA: cemmaorganizacion@gmail.com

IEO: Begona.SANTOS@efca.europa.eu

JNCC: tim.dunn@jncc.gov.uk

NEFSC: jason.roberts@duke.edu

PELAGIS: pelagis@univ-lr.fr

SEAPODYM: plehodey@groupcls.com

SMRU: psh2@st-andrews.ac.uk

 4. We note that Appendix S2, S4 and Figures 1 and 4 in your submission contain map/satellite images which may be copyrighted.

ANSWER: All captions and figures were completed with the source of the base map (https://www.gebco.net/).

Reviewer 1 # 

The article entitled "Towards a better characterisation 1 of deep-diving whales’ distributions by using prey distribution model outputs" aimed at comparing 2 habitat modelling approaches to predict the distribution of deep divers using surface environmental variables vs. low and mid-trophic levels outputs. The authors showed that including SEAPODYM covariates does not improve the model performance and conversely lead to drastic different predictions, raising serious concern regarding the reliability of the model output and the algorithm used (classical additive model). Despite the efforts of the authors to provide a clear methodology, the results lack of clarity (numerous repetitions, the results are not sufficiently discussed in regards to other studies) and the bias / limitations of the modelling procedure are not well discussed. I suggested several improvements, including testing other algorithms from machine learning and adding other covariates into the model as well as additional literature. The authors have a dataset will a great potential, and I'm convinced once the methodology will be made more robust and the results clearer, this article will be of great contribution to the habitat modelling and marine mammals’ communities!

ANSWER: We thank reviewer 1 for his careful reading of our manuscript and for pointing out methodological and redactional concerns. The methodology we used for the selection of variables in the models has been revised in the new version of the manuscript and a new metric has been used to evaluate the model performance in order to better meet the objectives of the study but we have not tested other algorithms as suggested. Indeed, generalized additive models have repeatedly shown their effectiveness to model cetacean distributions (e.g. Roberts et al., 2016; Becker et al., 2017; Cañadas et al., 2018; Virgili et al., 2019) and we believe they can also be applied in this study. Reviewer 1 proposes to use machine learning algorithms instead which is justified if the main objective is to predict the distribution of species without necessarily explaining this distribution (Gobeyn et al., 2019) but here our objective was primarily to determine the explanatory performance of models that use different types of explanatory variables. It therefore seemed more relevant to use statistical models rather than machine learning. In addition, surface environmental variables are commonly used in generalized additive models to model the distribution of deep-divers and we wanted to assess whether the addition of prey distribution variables provided additional information to explain their distribution. Therefore, a comparable modelling approach was required. In our future studies, we will consider machine learning algorithms to predict the distribution of species. We have clarified the results and removed repetitions to meet reviewer’s expectation (lines 304-366). We have more discussed bias and limitations of our study (lines 382-430, 549-557) and added more literature to discuss our study. We think the revised manuscript has improved as a result. 

Specific comments: 

Page 2: This result is surprising and not very intuitive. look forward reading some explanations about this; same. isn't it the opposite?; I am wondering how these prey field variables are not strongly autocorrelated, but I presume you have tested that?; more than a description of the most significant variables selected for each model, I would have expected some performance metrics (e.g. deviance explained) to compare the 3 models. I would have been much more informative and less descriptive to have such results in the abstract

ANSWER: We have changed the temporal resolution of the SEAPODYM variables (from 29 to 7 days) and fitted new models so the results have changed in this new version of the manuscript and we have reworked the abstract as a whole, which we hope will address the comments of reviewer 1.

Line 66: predictors? to avoid repetition

ANSWER: Changed

Line 77: also because such micronekton outputs are closer to trophic level of apex predators prey

ANSWER: Complement added in the text lines 77-78. 

Line 86: you could also add this reference which shows the use of SEAPODYM to relate olive ridleys turtles to productive eddies: https://www.sciencedirect.com/science/article/abs/pii/S0079661115300094

ANSWER: Reference added line 87. 

Lines 96-97: no need here to describe all the predictors you plan to use

ANSWER: Text deleted. 

Line 129: the resolution of the figures is very low. I assume it is due to the requested format by the journal, but figures in vector format such as pdf would be needed to get a better result

ANSWER: Figures were provided in a format requested by the journal; the poor resolution is probably due to the creation of the pdf submission.

Line 159: dynamic or oceanographic? to be consistent pick one of these 2

ANSWER: We have changed dynamic into oceanographic in the whole text. 

Line 172: you could have done the same procedure with SSH to detect ocean fronts and particularly eddies. SSH and SST fronts can be slightly different and might provide complementary variables. See Chambault et al 2017 DSR II for more information.

ANSWER: We thank the reviewer for this proposal, indeed, it is a variable that could be interesting. To our knowledge, this variable is not known to have an effect on the deep-diver distribution so we have chosen not to include it in the new version of the manuscript, we have limited ourselves to classically used variables but we will test this variable in our future studies. 

Line 174: to be consistent I would suggest to use the same products as the ones assimilated into SEAPODYM. I might be wrong but I think SEAPODYM assimilates SST and currents from CMEMS directly (GLORYS product). If this is the case I would recommend to use products from CMEMS rather than many different sources of data

ANSWER: We do not believe that it makes sense to use the same data source for the environmental and SEAPODYM variables as this will necessarily lead to a correlation between the variables and therefore we would not be able to obtain a combination of the environmental and SEAPODYM variables in the COMBINED models.

Line 176: from AVISO there is also a useful product assembling all the eddies with their associated characteristics (radius, type, amplitude...): it is called the eddy Atlas. It could provide another predictor, especially since your whales seem to target the Gulf Stream

ANSWER: We thank reviewer 1 for this information but we do not think this type of variable at a very fine spatial resolution is very relevant for visual observation data, it would be more relevant for tracking data and therefore would not be applicable here. 

Line 178: another dynamic variable that could be of great interest for such diving predators is the bottom temperature available on CMEMS. for these animals that dive to great depths for long periods it makes more sense to also look at vertical variables like bottom temp

ANSWER: We fully agree with reviewer 1 and the use of depth variables will be the subject of another publication. Here, the objective was to compare the SEAPODYM variables to environmental variables classically used in habitat models such as static variables and surface variables that was why we did not include bottom temperature in the models.

Line 242: you should also test the correlation using the vif function from the usdm package. the common way is to retain all variables with a vif coefficient < 4

ANSWER: As suggested by reviewer 1, we have tested the correlation with the VIF function, the results were almost identical as the correlation matrices calculated with this method and our method were identical within 0.001. In addition, our method allowed to test all combinations of variables without first selecting one variable among two correlated variables, contrary to the VIF function, which seemed more relevant, so we have kept our variable selection method.

Line 243: I think 4 is quite a low number of variables, and you therefore risk to strongly limit the performance of your model. In SDM, many studies commonly us more than 4 predictors, so I suggest you don't limit the number of variables and simply remove the ones that are not significant

ANSWER: We understand reviewer’s concern, we have constrained the number of variables in the models to a maximum of 4 (ensuring that they were all significant) to avoid excessive complexity of models and difficulty in their interpretation. Although this could limit the performance of our models, we were confident in the procedure parsimony. Indeed, the explained deviances obtained (between 20 and 40%) were high compared to other studies on cetaceans, suggestive of a good fit to the data, and for each model, only four or five variables were actually important, the others representing a very small percentage of the Akaike weights (S3.2 Fig. in Supplementary material 3). As a result, four variable models were the best compromise between avoidance of overfitting, reduction of complexity and maximisation of interpretability. We have added this paragraph lines 420-430 to discuss methodology we have employed. 

Line 245: I am very surprised by this result! especially because you consider for each layer both the biomass and production. could you please test with the vif function and see if it is still not correlated?

ANSWER: We have checked the correlations between the variables with our method and the VIF function and we confirm these variables were not correlated (see S3.1 Fig. in Supplementary material 3). 

Line 263: have you included the month and year as random effect or explanatory variable then?

ANSWER: We did not include month and year as random effect or explanatory variables in the models because as we discuss lines 408-519, “One of the problems faced when studying the distribution of deep-diving cetaceans through habitat modelling is the lack of sighting data, it is necessary to assemble datasets from different surveys to obtain sufficient data to fit habitat models. Sampling is therefore non-uniform, both spatially and temporally, with some areas, years or seasons being much more represented. For example, some years were only sampled in the northwest Atlantic area. Therefore, we chose to combine all the data to ensure we had enough data without considering a temporal effect in the models. However, temporality was taken into account when associating environmental conditions with effort segments, so when there was a sighting, it was associated with environmental conditions averaged over the week for the SEAPODYM variables and over the month for the environmental variables before the sighting, to take into account potential time-lags. When deep-diver data will be large enough to allow a more uniform temporal sampling, the temporal aspect will be taken into account in the models.”.

Line 296: to be more logical and consistent with the text I suggest to convert the bathymetry in absolute values the values on both axes are not readable the Y axis should not be in log?

ANSWER: As suggested by reviewer 1, we have changed the scale of the bathymetry in Figure 3. The Y axis is a relative density in individuals/750 km² so it has not to be in log. 

Line 302: I still have a big doubt about that. Thanks for sharing the correlation matrix though, I'll have a look at it, but I think running the vif is strongly recommended here

ANSWER: VIF also showed no correlation between the variables as shown in our correlation matrix in S3.1 Fig. 

Line 302: when reading Fig 3b it is the opposite, and much more logical indeed! maybe I missed sth here but your graph shows a positive relationship btw bathy biomass and whale's density

ANSWER: Results have changed in the new version of the manuscript because the temporal resolution of SEAPODYM variables have changed so we have modified the text here. 

Line 303: however, relationships with epi B and meso B seem very unclear. were these variables significant or you decided to retain 4 variables even if not significant? if you did so, I think this is not the proper way to do it as these 2 predictors seem to have a very little influence on your response variable

ANSWER: All variables selected in the final models were significant but may have less influence compared to other which may have a great influence, we do not believe the method should be questioned here. 

Line 304: in your legend you mention 2.5 to 97.5 percentile. Please be consistent

ANSWER: Reviewer 1 is absolutely right, it is a mistake on our part, we have made the necessary changes in the text.

Line 306: I don't really see that on your Fig 4b, probably due to the legend min max values. I suggest you adjust the colorbar to the values of each model, otherwise it is very hard to see any favourable habitat here

ANSWER: We agree the colorbar may not be fully optimised but we chose to apply the same colorbar to facilitate the comparison between the different models, on the advice of many colleagues. 

Line 319: I don't understand how your 97.5% percentile can be so tight towards the left-hand side for instance in b) bathy B? it means you have only very few data beyond 4?

ANSWER: We thank reviewer 1 for pointing out this mistake, we have modified the figure. 

Line 336: weird because you had no data on such a habitat. likely due to the influence of the slope but it raises serious concern about your model performance to predict outside they are of the training dataset. this needs to be discussed

ANSWER: We understand reviewer’s concern and we discuss this on lines 549-557 “We performed a gap analysis to highlight geographical extrapolation areas in which environmental and SEAPODYM variables were out ranges of surveyed conditions. The centre of the study area, near the Mid-Atlantic Ridge, was not surveyed but variable conditions remained within the ranges of surveyed conditions, allowing to predict distribution at a large scale. The gap analysis revealed large gaps in environmental space coverage across the study area, especially in the western part of the Atlantic Ocean for the ENVIRONMENTAL models and in tropical waters for the SEAPODYM models. This suggests that sampling effort was not sufficient in deeper and steeper areas and more intensive sampling effort performed in these areas could help better describe the habitat used by deep‐divers.”

Line 338: again I disagree with your interpretation here. if you consider the values between the 5 and 95% percentile, the highest densities are found for high values of Bathy B and Epi P, not the opposite

ANSWER: The results have changed here and so the text, we hope our new interpretations agree with reviewer’s interpretations. 

Line 342: the strong dissimilarities between the ENV and SEAPODYM predictions raise serious concerns regarding the reliability of your modelling approach

ANSWER: The dissimilarities between ENVIRONMENTAL and SEAPODYM predictions mainly raised the effect of the variables and this is likely due to the spatial resolution of the SEAPODYM variables as discussed lines 540-545 “Beaked and sperm whales seemed homogeneously distributed in oceanic waters, which was not very consistent with the sightings and highlighted the poorer performance of the models (higher RMSEs). As previously showed in Lambert et al. (2014), these smoother predictions may be due to the large resolution of the variables. The spatial resolution of the SEAPODYM variables is currently set to 0.25° but is planned to be refined to 0.125° and a finer spatial resolution may reduce this smooth effect.”.

Lines 364-368: already said earlier, not necessary here

ANSWER: Text removed. 

Lines 373-375: true but such approach could partly explain your relatively low deviances and such differences between your models for each species. For example, sperm whales in the Caribbean might adopt a totally different habitat selection compare to their conspecific in the North-East Atlantic, simply due to genetic differentiation or habitat available

ANSWER: We agree with reviewer 1, deep-divers may have different responses between north and northwest Atlantic and we discussed this on lines 394-407. However, explained deviances were not low compare to other studies that use visual observation data (Becker et al., 2017; Cañadas et al., 2018), which suggested a good fit to the data.

Line 385: another explanation could be related to the spatial resolution of your data. 0.25 degrees is quite coarse to depict specific associations between an animal and its environment. I have heard the SEAPODYM might be soon available daily at 0.08 degrees, so I think it is worth mentioning such limitation

ANSWER: We agree with reviewer 1 and we have added this on lines 443-444.

Line 387: 0.2% difference is very negligible to me so this is not a strong statement. For both species, the SEAPODYM model was less performant, so I don't see the point to include such mid-trophic level covariates

ANSWER: With the change in temporal resolution of the SEAPODYM variable, results have changed and the difference was less negligible but we agree with reviewer 1 that the SEAPODYM models were less performant than ENVIRONMENTAL and COMBINED models. 

Line 389: it would be interesting to incorporate more variables into each model and not limit it to 4 to see if it changes the results. if your SEAPODYM covariates appear to be important but less important than the 4 first other variables according to you AIC, it could be the reason why you did not find any SEAPODYM variables in the combined model for beaked whales. not sure it'll change the story but it is worth trying it

ANSWER: We agree with the reviewer’s comment but as shown in S3.2 Fig in Supplementary material 3, no SEAPODYM variables were important in the beaked whale COMBINED model and only production of epipelagic organisms was an important variable in the sperm whale COMBINED model so even if the number of variables was higher, SEAPODYM variables probably would not have been selected in the models or they would not have been significant.

Line 393: ok but the slope did not appear to be drive a lot beaked whale distribution neither... probably due to the coarse 0.25 ° resolution you used. A similar study predicting sperm whale distribution based on tracking data faced the same issue with the slope when incorporating it into the model: https://onlinelibrary.wiley.com/doi/full/10.1002/ece3.7154

ANSWER: We do not understand the reviewer’s comment here because slope appeared to be an important variable for beaked whales as shown in Fig 3 or in S3.2 Fig in Supplementary material 3. 

Line 395: there could also be a temporal mismatch (temporal lag) between SEAPODYM output and the growth rate of the actual prey of these species. This issue was discussed in Chambault et al 2017, DSR

ANSWER: We agree with reviewer’s comment and we have added this point on lines 456-457.

Page 17-18: 

ANSWER: As suggested by reviewer 1, we have removed repetitions on these two pages and the results have changed in the new version do it is difficult to answer to the specific comments written on these two pages.

Line 449: that's why I suggested to add the bottom temperature as another covariate. such variable was found to play a key role in sperm whale distribution around Mauritius: Chambault et al 2021, Ecol Evol

ANSWER: We fully agree and this will be done in another publication. 

Line 451: in theory maybe, but your results show the opposite! also sperm whales can forage on the bottom of the ocean and not necessarily conduct pelagic dives so you need to be careful with your statement: https://onlinelibrary.wiley.com/doi/full/10.1002/ece3.3322

ANSWER: We do not understand why our results show the opposite, the slope was not selected in the COMBINED model (Fig S3) and was not considered as important as shown in S3.2 Fig in Supplementary material, sperm whales are likely to feed at the bottom as well, but perhaps less consistently than beaked whales, which performed deeper dives in average.

Line 453: more than the euphotic depth, you should look at the mixed layer depth as well. for deep divers the light indication given by the Zeu might not be very informative while the MLD will provide information on the depth of the thermocline which is frequently used by diving animals. this variable can be easily found on Copernicus as well

ANSWER: We agree with reviewer 1; the mixed layer depth would probably be more informative for deep-divers because it provides information on the depth of the thermocline which is frequently used by diving animals and we will explore this in future studies. Here we used euphotic depth because it is a variable available in SEAPODYM but mixed layer depth would probably have been more relevant. We discuss this on lines 515-521. 

Line 461: repetition!

ANSWER: Text removed. 

Line 467: for animals that display a coastal pattern, like bowhead whales, bathymetry is indeed important. this is a general rule for all coastal animals while pelagic ones will describe more affinities for oceanographic structures like eddies, fronts. a similar relationship was found in narwhals which strongly use glaciers and fjords: https://link.springer.com/article/10.1007/s00300-018-2345-y

ANSWER: We agree with reviewer 1 and it is probably true for sperm whales but according to literature, beaked whales seem particularly linked to bathymetry and physiographic structures like canyons and seamounts.

Line 471: I strongly disagree! the maps were not similar at all between the ENV and SEAPODYM models. You can say that for beaked whales they were quite the same for the ENV and COMBINED models, but this is it

ANSWER: We have changed the text on lines 532-548 and better interpreted our results, we agree, it was poorly worded. 

Line 471: I would say totally different and you need to discuss that because it raises serious concern regarding your model performance and reliability. the choice of the algorithm might not be adequate. Machine learning algorithm offer a wide range of algo depicting complex relationships and might be appropriate and easy to implement on your data. Here, are few exmaples of recent studies involving machine learning in SDM:

Mannocci et al 2021

Chambault et al 2021, Ecol Evol

https://www.sciencedirect.com/science/article/abs/pii/S0304380018304010

https://www.koreascience.or.kr/article/JAKO201216636292901.page

https://link.springer.com/chapter/10.1007/978-1-4419-7390-0_8

ANSWER: We thank reviewer 1 for raising this point. As previously said, machine learning algorithms are justified if the main objective is to predict the distribution of species without necessarily explaining this distribution (Gobeyn et al., 2019) but here our objective was primarily to determine the explanatory performance of models that use different types of explanatory variables. It therefore seemed more relevant to use statistical models rather than machine learning. In addition, surface environmental variables are commonly used in generalized additive models to model the distribution of deep-divers and we wanted to assess whether the addition of prey distribution variables provided additional information to explain their distribution. Therefore, a comparable modelling approach was required. In our future studies, we will consider machine learning algorithms to predict the distribution of species. 

Page 20:

ANSWER: All the conclusion has been modified, we hope it will meet the reviewer’s expectations. 

Reviewer 2 # 

Virgili et al. have identified an important gap in our understanding of the link between potential habitat for deep-diving marine predators and the distribution of those predators- namely the influence of prey on the distribution of predators as an intermediate explanatory step between the environment and predators. The most important finding of this work is that the SEAPODYM model of pelagic fish distribution does not explain the distribution of deep-diving predators better than environmental data (except for some weak evidence for sperm whales).

While the science in the paper is basically sound (though see comment below regarding model derivation), the framing needs a bit of work before it should be published. That is, what is the driving hypothesis of this paper? There seem to be two possible frameworks:

1) The current hypothesis is that SEAPODYM variables better explain the distribution of beaked and sperm whales. This was found not to be the case (that is, this is a negative result). If this framework is preferred, it should be stated in the abstract, introduction and conclusion very clearly that SEAPODYM was ruled out as a good framework for describing the distribution of these deep divers

ANSWER: We thank the reviewer 2 for pointing out that the framework was not well enough defined, we fully agree. We have therefore modified the abstract (lines 46-48 " SEAPODYM outputs alone did not seem ideal for modelling prey of deep-diving cetaceans and maybe SEAPODYM was ruled out as a good framework for describing the species distribution, especially for beaked whales."), the introduction (lines 104-109 "We hypothesised that SEAPODYM variables would better explain the distribution of beaked whales and sperm whales because they are more proximal variables and because they characterise deep layers in which deep-divers feed. We also expected a better explanatory power of the model that combined the two types of variables because it considered static variables known to influence the beaked and sperm whale distribution, together with prey distribution data, presumably better suited to describe the species distributions.") and the whole conclusion (lines 577-593) to clarify the framework.

2) Given that, as stated in the discussion lines 396-424, SEAPODYM variables attempt to predict pelagic fish concentrations shallower than 1000m (that is, not the preferred prey of the study predators), a better test of SEAPODYM would be to choose a different predator to examine. Perhaps tuna or some other publicly available dataset (see e.g. https://portal.atn.ioos.us/) could be used to test the model more appropriately?

ANSWER: As stated in the discussion (lines 460-465), deep-divers can feed on prey larger than 20 cm at depths greater than 1,000 m and we agree we reviewer 2 that SEAPODYM variables would probably better explain the distribution of predator species such as tuna (which has been done). However, as stated in the introduction (lines 90-92 “Deep-divers are species of interest because they are sensitive to underwater noise pollution and an accurate knowledge of their distribution is crucial to mitigate the impact of human activities.”) and discussion (lines 374-378 “The models were fitted to sightings of deep-diving cetaceans (beaked and sperm whales) because their habitat use is still poorly known and they face important anthropogenic threats including activities producing high intensity noise (e.g. military sonars or seismic guns ), which makes them species of major interest.”), it is important to better understand the distribution of deep-divers as their distribution is poorly known and they are species sensitive to underwater noise pollution (e.g. active military sonars) so we need to identify new methods that could allow a better understanding of their habitats. The method with the SEAPODYM variables does not seem to be ideal for this according to our study but it allows us to progress in the process. 

Besides framing, the other major issue with the manuscript in its current form is that it does not appear that the SEAPODYM model has fundamentally different information than the ENVIRONMENTAL model, thus it is not surprising that it does not improve the explanatory power of the environmental model. That is, in the paragraph on line 160, the SEAPODYM model appears to derive from SST, bathymetric and surface current data, which is the same information that parameterizes the ENVIRONMENTAL model. Is not one then just a transformation of the other? What information does the SEAPODYM model have the that the environmental model does not. There may be something in how data was integrated over time, but it was not clear from the current manuscript how these models are substantially different. In the revision, please clarify how these models provide different information, given that they do not actually include information about prey in the SEAPODYM model.

ANSWER: We fully agree with reviewer 2, the description of the SEAPODYM variables was not complete enough and it was difficult to understand how these variables provide different information from environmental variables. We have therefore modified the section on SEAPODYM variables (lines 187-240) and we hope that the difference is clearer this time. SEAPODYM is a numerical model used to compute the spatial distribution of the biomass and the production of the micronekton and the zooplankton, in different layers of the water column. Energy transfers from the primary production to the groups of micronekton are parameterised in the model and a system of advection–diffusion–reaction equations, that take into account the vertical migrations of organisms, are used to model recruitment, ageing, mortality, and passive transport with horizontal currents. SEAPODYM makes it possible to obtain, in each cell of a grid, an estimate of the total quantity of organisms of each functional group present in each layer and of the productivity of the functional group in each layer.

Minor comments:

Abstract: please include a stronger summary in the abstract describing the main results (that SEAPODYM is not a good predictor of deep-diving cetacean habitat).

ANSWER: The part of the abstract concerning the results has been completely modified to better define the results we have obtained (lines 42-49 “For beaked whales, combining variables did not improve the model performance, no SEAPDOYM variables were selected in the best model, possibly because they prey on organisms less well predicted by SEAPODYM. For sperm whales, by combining variables, we slightly increased the sperm whale model performance (higher explained deviance and lower root mean squared error). SEAPODYM outputs alone did not seem ideal for modelling prey of deep-diving cetaceans and maybe SEAPODYM was ruled out as a good framework for describing the species distribution, especially for beaked whales. However, SEAPODYM gives new insights about potential prey fields targeted by deep-divers at large spatial scales.”). 

Line 75- also see croll et al. Croll, D. A., Marinovic, B., Benson, S., Chavez, F. P., Black, N., Ternullo, R., & Tershy, B. R. (2005). From wind to whales: trophic links in a coastal upwelling system. Marine Ecology Progress Series, 289, 117-130.

ANSWER: Reference included line 75.

Line 98- not clear what the difference between seapodym and seapodym model are?

ANSWER: ‘SEAPODYM model’ refers to the generalised additive model fitted to SEAPODYM variables while ‘SEAPODYM’ refers to the model of Lehodey et al. (2010) that simulated the prey distribution data. We have clarified this on lines 100-102 in the hope that it will be clearer.

Line 166- Why a 29-day sampling window? It seems like this may be appropriate for some variables but for others it would smooth over important information. A sensitivity analysis of choice of sampling window should be considered

ANSWER: We agree with reviewer 2, the 29-day sampling was not appropriate for SEAPODYM variables because the link between species distribution and prey distribution is probably more direct that the one with environmental variables. Consequently, all models were re-fitted with a 7-day temporal resolution for the SEAPODYM variables which was the resolution provided by the data provider.

General point of clarification- the spatial aspect of the modeling seems clear, but it would be helpful to have more explicit discussions of how the temporal aspect of the modeling was done, particularly for predator abundances that shift over time. Was there repeat sampling done or is predator abundance from a single survey? The “gapping” described appears to refer mostly to missing spatial data.

ANSWER: We thank the reviewer for pointing out this lack of clarification in the discussion. A new paragraph has been added to discuss this (lines 408-519 “One of the problems faced when studying the distribution of deep-diving cetaceans through habitat modelling is the lack of sighting data, it is necessary to assemble datasets from different surveys to obtain sufficient data to fit habitat models. Sampling is therefore non-uniform, both spatially and temporally, with some areas, years or seasons being much more represented. For example, some years were only sampled in the northwest Atlantic area. Therefore, we chose to combine all the data to ensure we had enough data without considering a temporal effect in the models. However, temporality was taken into account when associating environmental conditions with effort segments, so when there was a sighting, it was associated with environmental conditions averaged over the week for the SEAPODYM variables and over the month for the environmental variables before the sighting, to take into account potential time-lags. When deep-diver data will be large enough to allow a more uniform temporal sampling, the temporal aspect will be taken into account in the models.”).

Reviewer 3 # 

The main objective of this paper is to compare habitat models fit to different types of variables to for two deep diving cetaceans: sperm whales and beaked whales. The authors used generalized additive models (GAMs) to compare habitat models fit to 1) environmental variables, 2) simulated prey data from Spatial Ecosystem And POpulation DYnamics Model (SEAPODYM), and 3) combinations of the two variable types. The purpose of this study is to investigate if more direct measures of prey improve the fit of habitat models, when compared to prey proxies such as surface chlorophyll. The authors predict that more proximal variables, like prey biomass (simulated by SEAPODYM) will improve habitat models by reducing the time lag between phytoplankton biomass and the effect on higher trophic levels. The authors have presented and interesting and relevant study, however there are some issues with the methodology that must be addressed prior to publication. Major and minor concerns are detailed below.

Major Concerns

1. My first concern is within the ‘COMBINED’ habitat models for both species. The authors stated that they limited the computational burden within the combined model by only included a reduced number of variables; specifically, by only including variables selected within the final “ENVIRONMENTAL” and “SEAPODYM” models. This strikes me as a red flag and limitation. All relevant variables should be included in the ‘COMBINED’ models. If a variable is not significant in the “ENVIRONMENTAL” model, for example, this does not mean it will automatically not be significant in the “COMBINED” model. Specifically, an interaction between an environmental variable and a SEAPODYM variable could be significant in the “COMBINED” model, and this of course would not show up in the individual “ENVIRONMENTAL” or “SEAPODYM” models.

ANSWER: We understand the point raised by reviewer 3 and understand his concern. We have tried to fit the COMBINED models to all available variables but the computational burden was much too high because the area is very extended and the number of segments is very large (the computer crashed). However, we have revised the method to make it more robust and we hope this will meet the reviewer's expectations. Following Symonds & Moussalli (2011), we have determined the importance of each variable in the ENVIRONMENTAL and SEAPODYM models by summing the Akaike weights of the models in which the variable was selected and ranked all variables. We then included in the selection procedure of the COMBINED models all variables whose percentages of Akaike weight were greater than 25%, as these were the most important variables in the ENVIRONMENTAL and SEAPODYM models. With this method, all possible relevant variables were included in the COMBINED model and we can see in S3.2 Fig (Supplementary material S3) that a large number of variables included in the selection of the COMBINED model were not important so we think that the method is appropriate to answer our question. The new methodology was included lines 263-271. 

2. The authors do not reference/mention spatial auto-correlation or how it was accounted for within the models (e.g. include correlation structure in the model, bin the data at resolutions that are broader than the correlation).

ANSWER: We thank reviewer 3 for pointing this omission. Moran’s and Geary’s indexes have been calculated to ensure there was no spatial autocorrelation in the data using the ‘spdep’ R-package. We have added this specification lines 157-158. 

3. Explained deviance and AIC are great for model comparison and selection, but are insufficient in terms of model evaluation. AIC and explained deviance are what you use to choose your model, but doesn’t tell you anything about model performance. I would like to see an evaluation metric such as area under the curve (AUC) or true skill statistic (TSS). AUC indicates model performance, provide aggregate measures of performance across all possible classification thresholds, and allow performance of models to be compared. TSS measure the discriminatory ability of the SDMs (e.g. Becker et al. 2020).

ANSWER: Following reviewer’s recommendations, we have included a new parameter for model evaluation in addition to explained deviance which is commonly used to assess model performance (Becker et al., 2017). We have added the Root Mean Squared Error (RMSE) which measures the prediction errors and the model accuracy (the lower, the better; Neill & Hashemi, 2018). We did not used TSS or AUC as proposed by reviewer 3 because these parameters are used to assess the model performance of presence-only or presence-absence models not density models, which are the models we used. In this case, RMSE was more appropriate so we used it and we hope it meet the reviewer's expectations. 

4. A lesser concern is with the observations. As I am not a deep-diving whale expert, I am having trouble reconciling the use of surface observation for a study that focuses on fitting habitat models to sub-surface prey simulations. I am not contending that the use of surface observation is inappropriate, however I do think the methods would benefit from more explanation. For example, do we know how surface observation relates to deep dives for prey? Also including a couple references for studies that have used this type of observation for deep diving species’ habitat models previously would be useful.

ANSWER: We understand the reviewer’s concern, it may seem inappropriate to use prey simulations at depth to explain the distribution of surface sightings, it would be more appropriate to use tracking data. However, the accepted underlying assumption is that animals observed at the surface are present because they are mostly sensitive to the prey abundance and therefore a high prey biomass at depth could explain the presence of animals at the surface. Brodie et al. (2018) included two subsurface dynamic variables in species distribution models to describe the habitats of four pelagic species in the California Current System and increased the explanatory power and predictive performance of the models for most species. A more systematic use of depth variables could improve the tools available for the planning of human activities, especially for species that would be closely linked to processes at depth. We have added a part in the discussion about this, lines 383-393.

Minor comments

Line 57-59: “cetacean distributions are barely known” is an overgeneralization. If you are referring to deep-diving cetaceans, this needs to be made more explicit. Distributions of some cetaceans are actually quite well known and well modeled (e.g. Abrahms et al. 2019 – blue whales).

ANSWER: We agree with reviewer 3, we have specified “the deep-diving cetacean distributions” line 59.

Lines 291-294; 307, etc.: The way the authors refer to the “COMBINED” model for the beaked whale is confusing. The authors should state that the “COMBINED” model is identical to the “ENVIRONMENTAL” model at the beginning of the results section referencing beaked whales. Stating the comparison of AIC scores prior to making this point is confusing and makes it seem like these are different models (i.e. contain different variables). Similarly, within the Figure 3, it seems unnecessary to show identical maps for 3a and 3c. Instead, the authors should simply indicate that the “COMBINED” model is identical, and not in fact a separate model with separate results.

ANSWER: We have followed reviewer’s recommendation, we have simplified the text (lines 305-325) and we have removed the plot and the map referring to the beaked whale COMBINED model in figures 3 and 4. 

Line 333-334, etc.: Remove line “All selected variables were poorly correlated” lines from the results section. This is inherent based on the methodology. As such, you would only need to report if variables were slightly correlated, e.g.

ANSWER: Done. 

Line 418-424: I did find myself wondering the need to combine observation across such a large region. The authors offer an explanation in lines 418-424, but I don’t think this explanation is sufficient. Why not focus this study on one of the regions referenced? Would the authors expect prey data to be more relevant and improve fit at regionally specific scales? Is it possible that such a broad spatial area masks the impacts of prey data?

ANSWER: We thank reviewer 3 for raising this point. One of the problems faced when studying the distribution of deep-diving cetaceans through habitat modelling is the lack of sighting data, it is necessary to assemble datasets from different surveys and different areas to obtain sufficient data to fit habitat models. The combination of sighting data from different ecosystems (e.g. North-East and North-West Atlantic) may mask inter-regional differences in the relationships between cetacean densities and the environmental predictors and may mask the influence of prey data on species distribution. The objective here, was to compare models using environmental variables, prey distribution variables, and a combination of the two types of variables to determine the extent to which the models could be improved and not necessarily explain precisely the mechanisms influencing the distribution of beaked and sperm whales, which would be more consistent at the scale of a smaller region. We built basin-wide models and did not investigate region-specific models in order to leverage large sample sizes for investigating the explanatory power of SEAPODYM variables for habitat use of deep-divers. We acknowledge that pooling regions across an oceanic basin may introduce bias, but this study is a first investigation, and appreciate region-specific models would represent an obvious improvement (once enough data are available at the regional scale). We also think that in a global conservation context, it is important to study the broad scale cetacean distribution to obtain a more global prediction of this distribution and limit anthropogenic threats. We have added a paragraph to discuss this point lines 494-407. We thus agree with reviewer 3 it would be better to focus on one region, which is what we will intend to do in our further studies. 

Lines 483-484: Generally, in this part of the discussion, the authors continue to refer to the “COMBINED” models (plural) limiting the uncertainty within the interpolation. However, the authors previously state that the “COMBINED” model is identical to the “ENVIRONMENTAL” model for beaked whales, meaning that these conclusions would not be true for the beaked whales. The authors need to specifically reference the sperm whale example when making these statements/conclusions.

ANSWER: We agree with reviewer 3, we have changed the text through the whole discussion. 

References

Becker E.A, Forney K.A, Thayre B.J, Debich A.J, Campbell G.S, Whitaker K, et al. (2017). Habitat-based density models for three cetacean species off Southern California illustrate pronounced seasonal differences. Frontiers in Marine Science, 4:121.

Brodie S, Jacox M.G, Bograd S.J, Welch H, Dewar H, Scales K.L, et al. (2018). Integrating dynamic subsurface habitat metrics into species distribution models. Frontiers in Marine Science, 5: 219.

Cañadas A, de Soto N.A, Aissi M, Arcangeli A, Azzolin M, B-Nagy A., et al. 2018. The challenge of habitat modelling for threatened low density species using heterogeneous data: The case of Cuvier’s beaked whales in the Mediterranean. Ecological Indicators, 85: 128–136.

Gobeyn S, Mouton A.M, Cord A.F, Kaim A, Volk M, Goethals P.L. (2019). Evolutionary algorithms for species distribution modelling: A review in the context of machine learning. Ecological modelling, 392: 179-195.

Lambert C, Mannocci L, Lehodey P, Ridoux V. 2014. Predicting cetacean habitats from their energetic needs and the distribution of their prey in two contrasted tropical regions. PloS one, 9(8): e105958.

Lehodey P, Murtugudde R, Senina I. 2010. Bridging the gap from ocean models to population dynamics of large marine predators: a model of mid-trophic functional groups. Progress in Oceanography, 84(1): 69-84.

Neill S.P, & Hashemi M.R. (2018). Fundamentals of ocean renewable energy: generating electricity from the sea. Academic Press, pp 193-235.

Roberts J.J, Best B.D, Mannocci L, Fujioka E, Halpin P.N, Palka D.L, et al. 2016. Habitat-based cetacean density models for the U.S. Atlantic and Gulf of Mexico. Scientific Report, 6.

Symonds M.R, & Moussalli A. (2011). A brief guide to model selection, multimodel inference and model averaging in behavioural ecology using Akaike’s information criterion. Behavioral Ecology and Sociobiology, 65(1): 13-21.

Virgili A, Authier M, Boisseau O, Cañadas A, Claridge D, Cole T, et al. 2019. Combining multiple visual surveys to model the habitat of deep‐diving cetaceans at the basin scale. Global Ecology and Biogeography, 28(3): 300-314.

---

## [Decision Letter · Decision Letter 1]

7 Jun 2021

PONE-D-21-03978R1

Towards a better characterisation of deep-diving whales’ distributions by using prey distribution model outputs

PLOS ONE

Dear Dr. Virgili,

Thank you for submitting your manuscript to PLOS ONE. After careful consideration, we feel that it has merit but does not fully meet PLOS ONE’s publication criteria as it currently stands. Therefore, we invite you to submit a revised version of the manuscript that addresses the points raised during the review process.

We look forward to receiving your revised manuscript.

Kind regards,

Vitor Hugo Rodrigues Paiva, Ph.D.

Academic Editor

PLOS ONE

Journal Requirements:

Reviewers' comments:

Reviewer's Responses to Questions

**Comments to the Author**

1. If the authors have adequately addressed your comments raised in a previous round of review and you feel that this manuscript is now acceptable for publication, you may indicate that here to bypass the “Comments to the Author” section, enter your conflict of interest statement in the “Confidential to Editor” section, and submit your "Accept" recommendation.

Reviewer #1: All comments have been addressed

Reviewer #3: All comments have been addressed

2. Is the manuscript technically sound, and do the data support the conclusions?

Reviewer #1: Partly

Reviewer #3: Yes

3. Has the statistical analysis been performed appropriately and rigorously? 

Reviewer #1: Yes

Reviewer #3: Yes

4. Have the authors made all data underlying the findings in their manuscript fully available?

Reviewer #1: Yes

Reviewer #3: Yes

5. Is the manuscript presented in an intelligible fashion and written in standard English?

Reviewer #1: Yes

Reviewer #3: Yes

6. Review Comments to the Author

Reviewer #1: The authors have made a good job and undoubtfully improved the manuscript. However, it is unfortunate that the authors rejected most of my methodlogical suggestions and limited their approach to the "classical" way to run SDMs (i.e. limiting their number of covariates to 4, using GAMs only...). Especially given the growing community which is available in spatial modelling, novel techniques and approaches are emerging continuously, offering new perspectives and possibly more appropriate ways to use SDMs. The "common" way is not necessarily the best choice to make!

Some clarifications are still needed to justify the use of certain variables more than others to be scientifically valid.

I also have some concerns on the main finding of the paper stating that the addition of SEAPODYM variables to their more classical model improves the predictions and model performance. Considering the difference between the ENVIRONMENTAL and the COMBINED models is an order of agnitude of only 0.8, I am not convinced such a result is robust enough to be mentionned and it should rather be stated the other way to reject your main hypothesis. A negative result is still a result and not necessarly a meaningless one!

Reviewer #3: (No Response)

7. PLOS authors have the option to publish the peer review history of their article (what does this mean?). If published, this will include your full peer review and any attached files.

Reviewer #1: No

Reviewer #3: No

---

## [Author Response · Author response to Decision Letter 1]

13 Jul 2021

Dear Editor,

Thank you for considering our article entitled “Towards a better characterisation of deep-diving whales’ distributions by using prey distribution model outputs?”, for publication in the journal PLOS ONE. We have taken into account all comments from the editor and the reviewers regarding the article. 

We thank the reviewers for their careful reading of this new version of the manuscript and for pointing out elements in the abstract, methods and discussion that needed to be more clarified. We understand reviewer's comments about the way we run species distribution models and we believe we have clearly developed the limitations associated with our methodology in the discussion. The use of bottom variables is a very interesting proposal, especially for deep divers, and we have therefore mentioned it in the discussion. These variables are the subject of recent development and require further investigation before they can be used. This work is currently in project in our team and we feel is outside the scope of this study. The treatment of bottom variables or variables integrated in the water column is more complex than that of surface variables and it can also be criticised to use bottom variables to explain sightings recorded at the surface. The limitations associated with the use of bottom variables would be added to the limitations described in our study, and we believe it was better to separate the two approaches for greater understanding. We also understand the reviewer's caveats about limiting the number of variables in our models. Indeed, limiting the number of variables when your aim is solely to predict may be detrimental (but not necessarily, e.g. Authier et al. 2017), but for explanation it is not (e.g. Achen 2002, Shmuéli, 2010). Considering many variables in tandem will inevitably pose interpretation challenges, and may also lead to estimation problems of each individual effects because the information in the data is being spread thin on estimating a large number of parameters (see below for further explanation). We agree with the reviewer that an improvement in the models of only 0.8 was not sufficient and therefore we have modified the conclusions of our study rejecting our initial hypothesis. We hope this will meet the reviewer’s expectations.

We have fixed all typos and grammatical mistakes during the revision and ensured that the manuscript meets the PLOS ONE’s style requirements.

We hope that the revised manuscript addresses the different concerns raised by the editor and the reviewers, whom we thank for their careful reading and review of this manuscript. We feel the revised manuscript has improved as a result and hope it will be deemed acceptable for consideration of publication in PLOS ONE.

Sincerely,

On behalf of all co-authors,

Auriane Virgili

Concerning specific reviewer’s comments:

Reviewer 1 # 

Specific comments in the manuscript: 

Line 45: need reformulation

ANSWER: We have reformulated the sentence lines 43-45 “For beaked whales, no SEAPODYM variable was selected in the best model, possibly because SEAPODYM does not accurately simulate the organisms on which beaked whales feed on”.

Line 49: I'm wondering how seapodym could be ideal to predict the whale's distribution if it is "not ideal for modelling prey of deep-diving cetaceans"? your sentence needs to be rephrased to be consistent with your results.

ANSWER: We have rephrased the sentence lines 46-47 “SEAPODYM outputs were at best weakly correlated with sightings of deep-diving cetaceans, suggesting SEAPODYM may not accurately predict the prey fields of these taxa”.

Line 322: is it why you have one panel missing for BW? you mean your smooth curves and explained deviances were exactly the same for the ENV and COMBINED models? surprising...

ANSWER: As suggested by reviewer 3 in the previous version of the manuscript, we have pooled the results of the ENVIRONMENTAL and CONBINED models in the figures because the models were identical (same variables selected). 

Line 329: > 15°C seems already to be associated with higher densities, right?

ANSWER: That is right, we have changed the result. 

Lines 376, 384: again I found these SEAPODYM maps hard to interpret using the same colorbar as the 2 other models. I understand your concern of being able to compare the 3 models, but why don't you put the scale into log to have a better contrast? again hard to see from your maps

ANSWER: We have used a square root scale to have a better contrast because a log scale is not appropriate for low values. We hope it is better now. 

Line 454: again I found this argument very questionable, especially when using SDMs

ANSWER: Limiting the number of variables when your aim is solely to predict may be detrimental (but not necessarily, e.g. Authier et al., 2017), but for explanation it is not (see e.g. Achen, 2002). See Shmuéli (2010) for an indepth treatment of the difference between predictive modelling and explanatory modelling. Considering many variables in tandem will inevitably pose interpretation challenges, and may also lead to estimation problems of each individual effects (not to mention interactions) because the information in the data is being spread thin on estimating a large number of parameters. Another problem is the inevitable “all else being equal” assumption which result when individual effects are presented. These effects are conditional on all other variables in the model being set to their average values. Yet even slight covariations between variations makes this interpretation brittle as when one variable change, so do all the others (there by violating the underlying assumption of all-else-being-equal). One solution may be to present average predictive comparisons (Gelman & Pardoe, 2007) but these are more involved in their computations and not very much used (although they would be more accurate as they are marginal estimates, integrating over the other covariates in a model). The problem of having many variables in an explanatory model is that the geometry of the parameters space quickly become unwieldy and it is increasingly difficult to understand this geometry and interpret these parameters well. We are thus of the mind that for explanatory purposes, an appropriate strategy is to restrict the complexity. This may not be warranted when the goal is purely predictive, in which case interpretation is not sought either. In general, we should not expect a good predictive model to be also a good explanatory one (Shmuéli 2010, Smith & Santos, 2020). In the grey zone, a safety net is needed, and we have found the exposition in Achen (2002) to be quite persuasive. Out motivation for limiting the complexity to a maximum of 4 covariates can be traced back to ‘ART’: A Rule of Three. A statistical specification with more than three explanatory variables is meaningless (Achen, 2002, page 446). This seemingly outrageous statement is motivated by the empirical observation (which in our own experience is correct) that ‘With more than three independent variables, no one can do the careful data analysis to ensure that the model specification is accurate and that the assumptions fit as well as the researcher claims’ (Achen, 2002).

Line 643: I still found this sentence unclear and contradictory

ANSWER: We have rephrased the sentence lines 592-593 “SEAPODYM did not seem to model accurately the prey of deep-diving cetaceans assuming the later are sighted at the surface of their foraging grounds”. 

Line 644: by 0.8 !! this result is highly negligible so I won't mention it

ANSWER: We agree with the reviewer, we have changed the sentence lines 593-595 “By combining SEAPODYM and environmental variables the increase model performance for sperm whale was only marginal”. 

Line 655: given your main findings I'm not sure SEAPODYM will be very helpful to predict the distribution of such species

ANSWER: We agree with the reviewer, we have changed the sentence lines 602-605 “A more systematic use of SEAPODYM could allow to better define the limits of its use and a development of the model that would simulate larger prey beyond 1,000 m would probably better characterise the prey of deep-diving cetaceans”.

Specific comments in the rebuttal letter: 

Here we rewrite the response we have made in the rebuttal letter, we add the reviewer's comment about that response and we respond to the comment.

Rebuttal letter: We do not believe that it makes sense to use the same data source for the environmental and SEAPODYM variables as this will necessarily lead to a correlation between the variables and therefore we would not be able to obtain a combination of the environmental and SEAPODYM variables in the COMBINED models.

Reviewer’s comment: I disagree. Nowadays there are plenty of ocean products available from different databases and these can differ from one another. If SEAPODYM assimilates sst and currents data from one source, it does not necessarily mean that these 2 variables will be strongly correlated to your COMBINED model because complex calculations are done to produce mid-trophic level outputs. But if you're more familiar with AVISO products fine, you should simply say so because your justification to use such products is not valid

ANSWER: We are indeed more familiar with AVISO products, we use them routinely and the possibilities offered by AVISO are very large. However, it would be possible that the variables used to simulate SEAPODYM variables would be correlated with our variables if we used the same source, we did not have the opportunity to test this.

Rebuttal letter: We thank reviewer 1 for this information but we do not think this type of variable at a very fine spatial resolution is very relevant for visual observation data, it would be more relevant for tracking data and therefore would not be applicable here. 

 Reviewer’s comment: The eddy atlas from AVISO is not at a finer resolution as the other products you are using since the eddies are estimated from SLA and possibly from ARGO floats, so it is not a matter of spatial resolution. Would be interesting to see if your whales select some particular eddy types, just a suggestion. At least you could add few lines to the Discussion?

ANSWER: We agree with the reviewer, we have added few lines to mention that lines 524-526 “It would be particularly interesting to characterise eddies (e.g. cyclonic or anticyclonic) to assess whether deep-divers preferentially select a certain type of eddy as it would appear that prey aggregating at eddies are different depending on their rotation cycles [78]”.

Rebuttal letter: We fully agree with reviewer 1 and the use of depth variables will be the subject of another publication. Here, the objective was to compare the SEAPODYM variables to environmental variables classically used in habitat models such as static variables and surface variables that was why we did not include bottom temperature in the models.

Reviewer’s comment: it is a pity you exclude so many variables of interest, especially for deep divers. Bottom temperature has been used in a recent publication on sperm whales (Chambault et al 2021, Eco Evol) and I don't believe it will be that complicated to rerun your models with this additional variable

ANSWER: As mentioned above, the use of bottom variables is very interesting, especially for deep divers, and we have therefore mentioned it in the discussion lines 396-402 and 520-526. These variables are the subject of recent development and require further investigation before they can be used. This work is currently in project in our team and we feel is outside the scope of this study. The treatment of bottom variables or variables integrated in the water column is more complex than that of surface variables and it can also be criticised to use bottom variables to explain sightings recorded at the surface. The limitations associated with the use of bottom variables would be added to the limitations described in our study, and we believe it was better to separate the two approaches for greater understanding. Contrary to what the reviewer thinks, it is tedious and very time-consuming to re-run the models with other variables because it is necessary to extract the variables over 18 years, to associate the variables with the effort segments (more than 244,000 segments) and to re-run the models, which requires several days per model, and even more if variables are added. This will therefore be the subject of another study.

Rebuttal letter: The dissimilarities between ENVIRONMENTAL and SEAPODYM predictions mainly raised the effect of the variables and this is likely due to the spatial resolution of the SEAPODYM variables as discussed lines 540-545 “Beaked and sperm whales seemed homogeneously distributed in oceanic waters, which was not very consistent with the sightings and highlighted the poorer performance of the models (higher RMSEs). As previously showed in Lambert et al. (2014), these smoother predictions may be due to the large resolution of the variables. The spatial resolution of the SEAPODYM variables is currently set to 0.25° but is planned to be refined to 0.125° and a finer spatial resolution may reduce this smooth effect.”.

Reviewer’s comment: 0.08 degrees daily and not 0.125°. Not convinced this has anything to do with the spatial resolution but more of the lack of sightings and/or not inappropriate covariates. I bet if you would conduct the same analysis with the new resolution of SEAPODYM the results would be identical, meaning the differences are not due to the spatial resolution but rather to the covariates themselves

ANSWER: We thank the reviewer for pointing out this mistake, we have changed the value in the text. We agree that SEAPODYM variables are certainly not suitable for deep divers as SEAPODYM does not simulate large enough prey and deep enough layers, however a finer resolution of the variables could perhaps allow for more structured predictions and not obtaining homogeneous predictions. It could also be due to the lack of data as suggested by the reviewer but in this case, we would have observed the same problem on the maps of the ENVIRONMENTAL and COMBINED models. 

Rebuttal letter: We agree with the reviewer’s comment but as shown in S3.2 Fig in Supplementary material 3, no SEAPODYM variables were important in the beaked whale COMBINED model and only production of epipelagic organisms was an important variable in the sperm whale COMBINED model so even if the number of variables was higher, SEAPODYM variables probably would not have been selected in the models or they would not have been significant.

Reviewer’s comment: in your Table SI 31, I don't understand your last column "delta AIC". It does not match the AIC column so please explain. In the 4th and 5th rows of the combined models, there ARE seapodym layers,and the AIC are very similar to teh best model so I'm not really convinced by your answer here

ANSWER: There was an error in the Appendix due to rounding, the correction has been made. However, this does not change the results. We refer here to figure S3.2 which ranks the variables according to their Akaike weights and we can see that in the COMBINED beaked whale model there are only 4 variables that stand out as important and no SEAPODYM variables and for sperm whales 5 variables stand out with only epipelagic production for the SEAPODYM variables so including additional variables would probably not change anything in the model results. 

Rebuttal letter: We do not understand why our results show the opposite, the slope was not selected in the COMBINED model (Fig S3) and was not considered as important as shown in S3.2 Fig in Supplementary material, sperm whales are likely to feed at the bottom as well, but perhaps less consistently than beaked whales, which performed deeper dives in average.

Reviewer’s comment: your results (at least the previous version) showed no improvement when incorporation seapodym variables, suggesting the ENV model was more performant, that's why to me your results show the opposite

ANSWER: The reviewer's comment is still unclear, as slope does not appear to be an important variable for sperm whales unlike beaked whales, as shown in Figure S3.2, we can reasonably assume that they are less slope dependent than beaked whales. 

Rebuttal letter: We agree with reviewer 1; the mixed layer depth would probably be more informative for deep-divers because it provides information on the depth of the thermocline which is frequently used by diving animals and we will explore this in future studies. Here we used euphotic depth because it is a variable available in SEAPODYM but mixed layer depth would probably have been more relevant. We discuss this on lines 515-521. 

Reviewer’s comment: rather than justifying in the text why you have not included this variable, it would have been much more robust and convincing to actually include it! The fact that Zeu is available from SEAPODYM is not a convincing justification since the MLD is easily available from CMEMS website and can be incorporated in your models the same way your you did with the seapodym variables

ANSWER: We agree with the reviewer but as mentioned above, it takes a long time to add a variable and the objective here was not to test all the variables available for deep divers but to compare with the SEAPODYM variables so indeed it would have been preferable to add the MLD but it was not a priority for the study, it was more meaningful to focus on the temporality of the variables at 7 days.

Rebuttal letter: We thank reviewer 1 for raising this point. As previously said, machine learning algorithms are justified if the main objective is to predict the distribution of species without necessarily explaining this distribution (Gobeyn et al., 2019) but here our objective was primarily to determine the explanatory performance of models that use different types of explanatory variables. It therefore seemed more relevant to use statistical models rather than machine learning. In addition, surface environmental variables are commonly used in generalized additive models to model the distribution of deep-divers and we wanted to assess whether the addition of prey distribution variables provided additional information to explain their distribution. Therefore, a comparable modelling approach was required. In our future studies, we will consider machine learning algorithms to predict the distribution of species. 

Reviewer’s comment: I agree if your scope is also to explain the relationships then the GAMs or GLMs are a better choice than ML. However, the justification of keeping surface variables only because this is the "common way" to do it is not scientifically appropriate

ANSWER: As mentioned above, we plan to use bottom variables in habitat models but these variables are the subject of recent development and require further investigation before they can be used. We do not consider that keeping only surface variables because they are commonly used is the most appropriate but it is a fact that many studies on deep divers use surface variables in habitat modelling (e.g. Roberts et al., 2016; Becker et al., 2017; Cañadas et al., 2018), the use of bottom variables is very recent. Our objective was therefore to propose new variables, the SEAPODYM variables, which are integrated in the water column and could have been appropriate variables for deep divers. It turned out that they were not and for this reason new work was undertaken to use other variables that would be more suitable, so this is another study and mixing the two approaches would have complicated the message considerably. 

References

Achen, C. H. (2002). Toward a new political methodology: Microfoundations and ART. Annual review of political science, 5(1), 423-450.

Authier, M., Saraux, C., & Péron, C. (2017). Variable selection and accurate predictions in habitat modelling: a shrinkage approach. Ecography, 40(4), 549-560.

Becker E.A, Forney K.A, Thayre B.J, Debich A.J, Campbell G.S, Whitaker K, et al. (2017). Habitat-based density models for three cetacean species off Southern California illustrate pronounced seasonal differences. Frontiers in Marine Science, 4:121.

Cañadas A, de Soto N.A, Aissi M, Arcangeli A, Azzolin M, B-Nagy A., et al. 2018. The challenge of habitat modelling for threatened low density species using heterogeneous data: The case of Cuvier’s beaked whales in the Mediterranean. Ecological Indicators, 85: 128–136.

Gelman, A., & Pardoe, I. (2007). 2. Average predictive comparisons for models with nonlinearity, interactions, and variance components. Sociological Methodology, 37(1), 23-51. 

Roberts J.J, Best B.D, Mannocci L, Fujioka E, Halpin P.N, Palka D.L, et al. 2016. Habitat-based cetacean density models for the U.S. Atlantic and Gulf of Mexico. Scientific Report, 6.

Shmueli, G. (2010). To explain or to predict? Statistical science, 25(3), 289-310.

Smith, A. B., & Santos, M. J. (2020). Testing the ability of species distribution models to infer variable importance. Ecography, 43(12), 1801-1813.

---

## [Editor Report · Decision Letter 2]

22 Jul 2021

Towards a better characterisation of deep-diving whales’ distributions by using prey distribution model outputs?

PONE-D-21-03978R2

Dear Dr. Virgili,

We’re pleased to inform you that your manuscript has been judged scientifically suitable for publication and will be formally accepted for publication once it meets all outstanding technical requirements.

Kind regards,

Vitor Hugo Rodrigues Paiva, Ph.D.

Academic Editor

PLOS ONE

---

## [Editor Report · Acceptance letter]

26 Jul 2021

PONE-D-21-03978R2 

Towards a better characterisation of deep-diving whales’ distributions by using prey distribution model outputs? 

Dear Dr. Virgili:

I'm pleased to inform you that your manuscript has been deemed suitable for publication in PLOS ONE. Congratulations! Your manuscript is now with our production department. 

Kind regards, 

on behalf of

Dr. Vitor Hugo Rodrigues Paiva 

Academic Editor

PLOS ONE